# COUNTING HALLUCINATIONS IN DIFFUSION MODELS

## ABSTRACT

Diffusion probabilistic models (DPMs) have demonstrated remarkable progress in generative tasks, such as image and video synthesis. However, they still often produce hallucinated samples (*hallucinations*) that conflict with real-world knowledge, such as generating an implausible duplicate cup floating beside another cup. Despite their prevalence, the lack of feasible methodologies for systematically quantifying such hallucinations hinders progress in addressing this challenge and obscures potential pathways for designing next-generation generative models under factual constraints. In this work, we bridge this gap by focusing on a specific form of hallucination, which we term **counting hallucination**, referring to the generation of an incorrect number of instances or structured objects, such as a hand image with six fingers, despite such patterns being absent from the training data. To this end, we construct a dataset suite **CountHalluSet**, with well-defined counting criteria, comprising ToyShape, SimObject, and RealHand. Using these datasets, we develop a standardized evaluation protocol for quantifying counting hallucinations, and systematically examine how different sampling conditions in DPMs, including solver type, ODE solver order, sampling steps, and initial noise, affect counting hallucination levels. Furthermore, we analyze their correlation with common evaluation metrics, such as Fréchet Inception Distance (FID), revealing widely-used perceptual metrics fail to capture counting hallucinations consistently.

## 1 INTRODUCTION

Diffusion probabilistic models (DPMs) (Ho et al., 2020; Dhariwal & Nichol, 2021; Rombach et al., 2022), commonly referred to as diffusion model, have achieved considerable success in generative modeling, producing high-quality samples across various tasks such as text-to-image generation (Ho & Salimans, 2021; Rombach et al., 2022), audio synthesis (Kong et al., 2021), video generation (Ho et al., 2022), 3D object generation (Luo & Hu, 2021; Vahdat et al., 2022), and protein structure prediction (Jumper et al., 2021; Abramson et al., 2024). Despite these impressive advancements, diffusion models still struggle to faithfully capturing factual properties from training data, often generating outputs that violate consistency with real-world knowledge. A notable example is the generation of objects with an incorrect number of spatially intricate components, such as extra human fingers (Narasimhaswamy et al., 2024) or additional animal limbs (Borji, 2023).

Recent studies have increasingly focused on identifying and analyzing common failure modes in text-guided diffusion models. For instance, Borji (2023) systematically concluded several qualitative shortcomings in text-to-image models. Additionally, Liu et al. (2024) proposed an adversarial search method to uncover undesirable behaviors and failure cases in text-to-image generation, while Wu et al. (2023) employed an iterative approach to discover and mitigate spurious correlations.

More recently, *hallucination*, a specific failure mode in diffusion models, has garnered attention in the research community (Aithal et al., 2024). Unlike previously discussed text-prompt-based failure modes, which can be evaluated by measuring the alignment between the input conditions and outputs (e.g., CLIP score (Hessel et al., 2021) and VQA score (Huang et al., 2025; Zarei et al., 2024)), or more obvious failure modes such as severe distortions, blur, and artifacts, hallucinated samples (*hallucinations*) often appear visually plausible, and emerge independently of the conditioning signal, manifesting as subtle violations of factual consistency with respect to the training data, such as incorrect object counts, geometries that appear plausible but are physically impossible, or other breaches of physical laws. These fine-grained distinctions between valid samples and hallucinations present a critical challenge for current AI systems, which often fail to detect them reliably. For

Figure 1: Example comparison between the proposed **CountHalluSet** used for training (left) and the corresponding **counting hallucinations** generated by the model (right). Each dataset contains different objects with specific counting criteria. For example, in ToyShape and SimObject, each image contains at most one instance per category and at least one instance overall, whereas in RealHand, each image contains exactly five fingers. Counting hallucinations refer to generated images that violate the dataset's counting rules, such as two apples in an image or an image containing no objects but a normal background, even though these patterns never appear in the SimObject dataset.

example, even state-of-the-art large vision-language models like GPT-4o still struggle with basic tasks like accurately counting the number of fingers in a clear hand image. To this end, establishing a standardized and well-defined protocol for reliably quantifying hallucinated samples is essential, as it provides a foundation for outlining pathways toward designing next-generation generative models with strict factual consistency, an ability that is critical for their role as world models (Ha & Schmidhuber, 2018; Ding et al., 2024; Bruce et al., 2024).

Among the various common types of hallucinated elements, such as counts, colors, spatial relationship, geometry, and physical realism, counting errors are particularly suitable for systematic study because they are not only easy to identify but also straightforward to quantify. This motivate us to begin by defining a specific form of hallucination termed **counting hallucination**. *In this setting, the number of objects in training data is explicitly predetermined. Consequently, a generated image is deemed hallucinated whenever its object counts deviate from these predefined values.* For example, an image depicting a hand with six fingers is classified as a counting hallucination, since such a pattern never occurs in the RealHand dataset, as shown in the last row of Fig. 1. To facilitate the study of such phenomena, we construct a dataset suite, CountHalluSet, consisting of three datasets that span a spectrum of morphological complexity for the countable objects: ToyShape (*triangle*, *square*, *pentagon*), SimObject (*mug*, *apple*, *clock*), and RealHand (*finger*). Figure 1 illustrates representative examples from these datasets alongside corresponding counting hallucinations.

In our quantification protocol for counting hallucinations, a critical step is the accurate identification and quantification of objects. To this end, we employ counting models tailored to each dataset. Leveraging these datasets and counting models, we examine how various denoising conditions (e.g., sampling steps, initial noise, the order of ODE solvers, and solver type) within diffusion models affect counting hallucination rates. Besides, we conduct correlation analysis between counting hallucination rate and perceptual metrics, including Fréchet Inception Distance (FID) Heusel et al. (2017) and MUSIQ Ke et al. (2021).

Several significant findings or contributions include:

- Commonly applied numerical strategies in diffusion models, such as increasing sampling steps from 25 to 100, often fail to mitigate counting hallucinations in generating human hands, even though these techniques are effective in simpler synthetic datasets.

- A weak correlation is observed between the counting hallucination rate and perceptual metrics, whereas a strong correlation emerges between the non-counting failure rate and these metrics.

- Based on one of our findings, we propose a novel and promising training paradigm, termed joint-diffusion models, which can significantly reduce counting-based hallucinations and non-counting failures in the RealHand dataset.

## 2 RELATED WORK

### 2.1 HALLUCINATIONS IN DIFFUSION MODELS

Inspired by the growing attention to hallucination in large language models (LLMs) (Kadavath et al., 2022; Ji et al., 2023; Huang et al., 2023), the study of hallucinations in unconditional diffusion models has also garnered increasing interest. In particular, Aithal et al. (2024) proposed a qualitative hypothesis suggesting that hallucinated samples may arise from data interpolating between distinct modes. Despite this insight into potential causes, a systematic quantitative analysis of hallucinations in diffusion models remains lacking.

This motivates us to facilitate the accurate identification and quantification of hallucinations in diffusion-generated outputs. Such an investigation can deepen our understanding of undesirable behaviors exhibited by these models, and clarify how the magnitude of denoising errors, such as variations in sampling steps or the order of ODE solvers, affects the severity of hallucinations.

### 2.2 QUANTITATIVE METRICS IN IMAGE GENERATION

The Visual Question Answering (VQA) score (Huang et al., 2025; Zarei et al., 2024) and the CLIP score (Hessel et al., 2021) are two widely used metrics for evaluating the performance of text-guided generated images. However, such evaluation methods that rely on language-based models are inherently limited when applied to assess hallucinated images. One line of evidence is the statistical lower bound on the propensity of pretrained language models to hallucinate certain fact types, which constrains the upper bound of detection accuracy based solely on language models (Kalai & Vempala, 2024). Another indication is the persistent performance gap between human annotators and LLMs in classifying hallucinated textual content (Lin et al., 2022; Li et al., 2023).

In addition, several widely used quantitative evaluation metrics, such as FID (Heusel et al., 2017), IS (Salimans et al., 2016), and Precision & Recall (Sajjadi et al., 2018), and MUSIQ (Ke et al., 2021) are commonly employed in image generation tasks. These metrics rely on the feature embeddings extracted from a pretrained Inception-v3 model (Szegedy et al., 2016). However, *whether these metrics can capture the severity of hallucinations in generated datasets* remains an open question. In this work, we aim to address this gap from the perspective of counting hallucinations.

## 3 PRELIMINARY: DIFFUSION PROBABILISTIC MODELS

### 3.1 FORWARD (DIFFUSION) PROCESS

**Forward Process.** Given the observed data $x_0 \sim q_0(x_0)$, Diffusion Probabilistic Models (DPMs) (Sohl-Dickstein et al., 2015; Ho et al., 2020) introduce a *forward process* (a.k.a, *diffusion process*) that gradually perturbs the data by adding Gaussian noise over $T$ discrete timesteps. At at each timestep $t$, the marginal distribution of the noisy sample $x_t$ conditioned on the original data $x_0$ follows:

$$q_{0:t}(x_t|x_0) = \mathcal{N}(x_t \mid \sqrt{\bar{\alpha}_t}x_0, (1 - \bar{\alpha}_t)I), \quad \alpha_t := 1 - \beta_t, \ \bar{\alpha}_t := \prod_{j=1}^{t} \alpha_j, \quad (1)$$

where $t \in \{0, 1, \ldots, T\}$ and $\{\beta_i\}_{i=1}^{T}$ is a pre-defined noise schedule dependent on $t$, commonly chosen to be linear (Ho et al., 2020) or cosine (Nichol & Dhariwal, 2021) schedules. The factor $\bar{\alpha}_t$ controls the signal-to-noise ratio at step $t$: $\bar{\alpha}_0 = 1$ (no noise) and $\bar{\alpha}_T \approx 0$ (nearly pure noise). An equivalent formulation of Eq. (1) is:

$$x_t = \sqrt{\bar{\alpha}_t}x_0 + (1 - \bar{\alpha}_t)\epsilon, \quad \epsilon \sim \mathcal{N}(0, I). \quad (2)$$

As $T$ becomes sufficiently large, the marginal distribution $q_{0:T}(x_T|x_0)$ approaches a standard Gaussian distribution $\mathcal{N}(0, I)$. In widely-used DPMs (Ho et al., 2020; Nichol & Dhariwal, 2021), $T$ is typically set to 1000.

### 3.2 REVERSE (DENOISING) PROCESS.

**Training Objective.** The *reverse process* (a.k.a., *denoising process*) of DPMs aims to iteratively recover the observed data based on the diffused data $x_T$. Specifically, DPMs attempt to predict the

noise $\boldsymbol{\epsilon}$ from the data $\boldsymbol{x}_t$ by using a neural network $\boldsymbol{\epsilon}_\theta(\boldsymbol{x}_t, t)$, known as *noise prediction model*. The parameter $\theta$ is optimized by minimizing the following objective (Ho et al., 2020; Song et al., 2022):

$$\min_\theta \mathbb{E}_{t, \boldsymbol{x}_0, \boldsymbol{\epsilon}} [\pi(t) \|\boldsymbol{\epsilon}_\theta(\boldsymbol{x}_t, t) - \boldsymbol{\epsilon}\|_2^2], \tag{3}$$

where $\pi(t) > 0$ is a weighting function.

**Ancestral Sampling.** Given noisy data $\boldsymbol{x}_T$, samples from the original distribution $q_0(\boldsymbol{x}_0)$ can be obtained by reversing the diffusion process with the same number of steps, commonly referred to as *ancestral sampling*:

$$p_\theta(\boldsymbol{x}_{t-1} \mid \boldsymbol{x}_t) = \mathcal{N}(\boldsymbol{x}_{t-1} \mid \frac{1}{\sqrt{1 - \beta_t}} (\boldsymbol{x}_t + \beta_t \nabla_{\boldsymbol{x}_t} log\, q_t(\boldsymbol{x}_t)), \beta_t \boldsymbol{I}), \tag{4}$$

where $\nabla_{\boldsymbol{x}_t} log\, q_t(\boldsymbol{x}_t)$ is the only unknown term, referred as *score function* (Song et al., 2022).

In practice, DPMs approximate the scaled score function $-\sqrt{(1 - \bar{\alpha}_t)} \nabla_{\boldsymbol{x}_t} log\, q_t(\boldsymbol{x}_t)$ via the noise prediction model. Therefore, an equivalent formulation of Eq. (3) is:

$$\boldsymbol{x}_{t-1} = \boldsymbol{\mu}_\theta(\boldsymbol{x}_t, t) + \sqrt{\beta_t}\boldsymbol{\epsilon}, \quad \boldsymbol{\mu}_\theta(\boldsymbol{x}_t, t) = \frac{1}{\sqrt{1 - \beta_t}} \boldsymbol{x}_t - \frac{\beta_t}{\sqrt{1 - \beta_t}} \frac{1}{\sqrt{1 - \bar{\alpha}_t}} \boldsymbol{\epsilon}_\theta(\boldsymbol{x}_t, t). \tag{5}$$

**ODE-Based Accelerated Sampling.** The ancestral sampling in Eq. (4) contains a stochastic term (Kloeden et al., 1992), while there is an associated deterministic process, which can be formulated as an ordinary differential equation (ODE) (a.k.a, *probability flow ODE* (Song et al., 2022)):

$$\frac{\mathrm{d}\boldsymbol{x}(\tau)}{\mathrm{d}\tau} = f(\tau)\boldsymbol{x}(\tau) + h(\boldsymbol{x}(\tau), \tau), \quad f(\tau) = -\frac{1}{2}\beta(\tau), \ h(\boldsymbol{x}(\tau), \tau) = -\frac{1}{2}\beta(\tau)\nabla_{\boldsymbol{x}(\tau)} log\, q_\tau(\boldsymbol{x}(\tau)), \tag{6}$$

where $\tau \in [T, 0]$. Let $\boldsymbol{x}(\tau)$ denote the ideal trajectory that satisfies this ODE with the true score function $h(\boldsymbol{x}(\tau), \tau)$ and starts from an ideal diffused data $\boldsymbol{x}(T)$ drawn from the true diffused distribution $q_T(\boldsymbol{x}(T))$. Set the discrete time steps as $\tau_k = T - k\Delta t$ for $k = 0, 1, \ldots, N$, where $\tau_N = 0$ Applying the variation-of-constants formula for the exact solver over one step:

$$\boldsymbol{x}(\tau_{k+1}) = \mathcal{G}(\tau_{k+1}, \tau_k)\boldsymbol{x}(\tau_k) + \int_{\tau_k}^{\tau_{k+1}} \mathcal{G}(\tau_{k+1}, u)h(\boldsymbol{x}(u), u)\mathrm{d}u \tag{7}$$

where $\mathcal{G}(t_2, t_1) = e^{\int_{t_1}^{t_2} f(v)\mathrm{d}v}$ is the state transition operator for the linear part $\frac{\mathrm{d}\boldsymbol{z}}{\mathrm{d}\tau} = f(\tau)\boldsymbol{z}$. $\Delta t > 0$ denotes the step size, allowing ODE-based solvers to accelerate the sampling process with larger step sizes compared to ancestral sampling. The total number of sampling steps is given by $T/\Delta t$. Since the linear part $\mathcal{G}(t_2, t_1)$ can be computed in closed form, most accelerated ODE-based samplers (Song et al., 2021; Liu et al., 2022; Zhang & Chen, 2023; Lu et al., 2022; 2023) devote their numerical efforts, such as higher solver orders and more sampling steps, to accurately approximating the nonlinear integral term involving $h$. More theoretical analysis can be found in Appendix A.1 to A.3.

# 4 FORMAL DEFINITION, DATASETS, AND EVALUATION PROTOCOL FOR COUNTING HALLUCINATIONS

In this section, we describe the methodology and datasets for quantifying counting hallucinations. We first define counting criteria and the associated counting hallucinations, then introduce the datasets in the CountHalluSet suite. For each dataset, we specify its counting criteria and corresponding counting hallucinations, and outline the standardized protocol used for their quantification.

## 4.1 COUNTING HALLUCINATIONS: DEVIATIONS FROM THE COUNTING CRITERIA

In this section, we provide a general definition of counting criteria within training datasets and define counting hallucinations as deviations from these predefined constraints.

**Counting criteria in a training dataset**. Let $\mathcal{D}_{\text{ref}}$ be a training dataset. For any reference sample $\boldsymbol{x} \in \mathcal{D}_{\text{ref}}$, the number of objects of category $c$ in $\boldsymbol{x}$, $N_c(\boldsymbol{x})$, are given by a predefined set $\mathcal{S}_c \subseteq \mathbb{Z}_{\geq 0}$ (e.g., $\{0, 1\}$ or $\{1, 2, 3\}$), i.e., $N_c(\boldsymbol{x}) \in \mathcal{S}_c$. We further require that a valid reference sample contains at least one object, i.e. $\sum_{c \in C} N_c(\boldsymbol{x}) \geq 1$, where $C$ is the set of considered categories.

**Counting hallucinations as deviations from the counting criteria**. Given a generated sample $\hat{\boldsymbol{x}}$, we say $\hat{\boldsymbol{x}}$ exhibits a *counting hallucination* with respect to $\mathcal{D}_{\text{ref}}$ if both of the following conditions hold:

1. **Sufficient visual quality for counting.** The image $\hat{x}$ should exhibit sufficient visual quality such that the background and any present objects are clearly discernible. This ensures that counting hallucinations can be distinguished from other prominent failure modes, such as severe degradations. We define a binary indicator $\mathbb{I}_{\mathrm{CRI}}(\hat{x}) \in \{0, 1\}$, referred to as the *counting-ready indicator*, where 1 denotes that the image quality is sufficient for counting, and 0 indicates otherwise. In practice this indicator can be determined via human annotation, by a classifier calibrated against human annotations, or by other automated criteria.

2. **Violation of counting facts.** A violation of counting facts occurs if there exists a category $c$ such that $N_c(\hat{x}) \notin \mathcal{S}_c$, or if the image contains no objects (i.e., $\sum_{c \in C} N_c(\hat{x}) = 0$) as the dataset requires at least one object per sample. In practice, we use a counting model to predict object counts in generated images as evidence for such violations.

A counting hallucination (CH) is identified if its quality is sufficient for counting ($\mathbb{I}_{\mathrm{CRI}}(\hat{x}) = 1$) and the predicted counts deviate from the counting criteria ($\exists c : N_c(\hat{x}) \notin \mathcal{S}_c$ or $\sum_{c \in C} N_c(\hat{x}) = 0$). Formally,

$$\mathbb{I}_{\mathrm{CH}}(\hat{x}) = \mathbb{I}_{\mathrm{CRI}}(\hat{x}) \wedge \Big( \exists c : N_c(\hat{x}) \notin \mathcal{S}_c \ \vee \ \sum_{c \in C} N_c(\hat{x}) = 0 \Big).$$

### 4.2 DATASETS, COUNTING HALLUCINATIONS AND EVALUATION PROTOCOL

The CountHalluset suite consists of three datasets for studying counting hallucinations: ToyShape, SimObject, and RealHand. Examples from each dataset and corresponding counting hallucinations are shown in Fig. A1 and A2. Additional statistical summaries are provided in Appendix A.4.

#### 4.2.1 TOYSHAPE

**Dataset description**. The ToyShape dataset comprises of 30,000 images with three geometric shapes: *triangle*, *square*, *pentagon*. All shapes are white, with an equal area of 120 pixels, and no shapes overlap within any image. The counting criteria specify that each image contains at most one instance of each shape category and at least one shape (in total), as shown in the top-left of Fig. 1. Formally, we have $C = \{triangle, square, pentagon\}$. For each class $c \in C$, we set $N_c(x) \in S_c = \{0, 1\}$, subject to the constraint $\sum_{c \in C} N_c(x) \geq 1$.

**Counting hallucinations**. According the counting criteria defined above, any image generated by the model trained on ToyShape that contains two or more shapes of the same category is considered a counting hallucination, such as an image containing two pentagons in the top-right region of Fig. 1. Besides, empty images (i.e., containing only the black background without any shapes) are also classified as counting hallucinations. Examples can be found in Appendix A.5.

**Counting model used in ToyShape**. We utilize a counting model to identify counting errors within images, without using a counting-ready indicator, as the dataset's simplicity makes severely degraded images rare. In specific, we fine-tune a ResNet-50 (He et al., 2016) using over 400k ToyShape data, with each category appearing 0–3 times per sample (i.e., $S_c = \{0, 1, 2, 3\}$), aiming to capture a wide range of plausible scenarios, including counting-valid samples and counting hallucinations, enabling accurate quantification. The model can achieve over 99.9% counting accuracy on generated data.

**Quantification of counting hallucinations**. Given a diffusion model trained on the ToyShape dataset, we generate a set of samples equal in size to the training dataset (30,000 images). We then employ the counting model to predict the number of objects in each generated image. Lastly, images that violate the counting criteria are identified as counting hallucinations, as illustrated in Fig. 2.

#### 4.2.2 SIMOBJECT

**Dataset description**. The SimObject dataset includes 30,000 rendered images of everyday objects. It comprises three object categories: *mug*, *apple*, *clock*, each with 10 distinct intra-class variations. All objects are randomly placed on a wooden tabletop under fixed lighting conditions, with minimal mutual occlusion if any, as demonstrated in the middle of the left side of Fig. 1. The dataset is synthetically generated in Unreal Engine 5 (Epic Games, 2024), featuring photorealistic rendering with accurate simulation of object geometry, material attributes, lighting, and reflections. The counting criteria for this dataset are similar to those of the ToyShape dataset (see Sec. 4.2.1). Formally, we

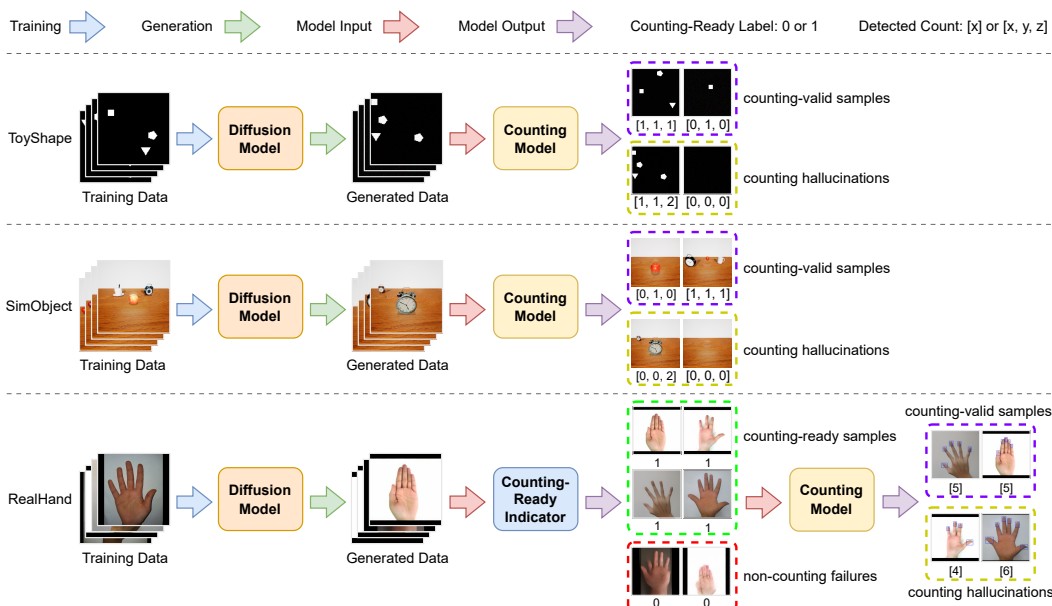

Figure 2: Evaluation procedure in the CountHalluSet suite. For ToyShape and SimObject, generated images are directly assessed for counting. For RealHand, a counting-ready indicator (a binary classifier aligned with human judgments) is introduced to separate counting hallucinations from other non-counting failures, such as severely deformed fingers. Count labels correspond to ToyShape: *triangle*, *square*, *pentagon*; SimObject: *mug*, *apple*, *clock*; RealHand: *finger*.

have $C = \{mug, apple, clock\}$. For each class $c \in C$, we set $N_c(\boldsymbol{x}) \in S_c = \{0, 1\}$, subject to the constraint $\sum_{c \in C} N_c(\boldsymbol{x}) \geq 1$.

**Quantification of counting hallucinations**. The definition of counting hallucinations, the construction of the counting model, and the quantification protocol follow the same procedure as in the ToyShape dataset, owing to the similarity of their counting criteria, as shown in Fig. 2.

### 4.2.3 REALHAND

**Dataset description**. To investigate counting hallucinations in real-world scenarios, we construct the RealHand dataset comprising 5,050 human hand images sourced from the 11k Hands dataset (Afifi, 2019), Kaggle[1], and Roboflow[2]. Specifically, we aggregate raw images from these sources and remove those with blur or artifacts. Each image depicts a human hand with five fingers, capturing both dorsal (back) and palmar (front) views under varying lighting conditions and backgrounds (e.g., white surfaces, blackboards, walls), as shown in the lower-left of Fig. 1. The counting criteria specify that each image contains exactly five fingers, i.e., $C = \{finger\}$ with $N_c(\boldsymbol{x}) \in S_c = 5$ for each $c \in C$, and $\sum_{c \in C} N_c(\boldsymbol{x}) = 5$.

**Counting hallucinations and non-counting failures**. Generating realistic hand images is inherently prone to unexpected failures beyond counting hallucinations (e.g., extra or missing fingers), such as blurriness, severely deformed fingers, and visual artifacts. Therefore, we use a counting-ready indicator (i.e., a binary classifier) to identify these cases. Examples of counting hallucinations and non-counting failure samples can be found in Appendix A.7.

**Counting-ready indicator and counting model used in RealHand**. To separate counting hallucinations from previously mentioned failure modes, we finetune a MaxViT (Tu et al., 2022) model on 2.5k annotated generated images to serve as a counting-ready indicator aligned with human judgments. For counting, we finetune a YOLO-12 (Tian et al., 2025) model on 2k fingertip-annotated generated images. Based on human evaluation of 500 randomly sampled images by three annotators,

---

[1] https://www.kaggle.com/
[2] https://universe.roboflow.com/

the indicator and counting model achieve over 96% and 99% accuracy, respectively. Examples of YOLO detection results can be found in Appendix A.6.

**Quantification of counting hallucinations**. Unlike the ToyShape and SimObject datasets, where all generated images are directly evaluated due to their simplicity and the rarity of severe failure cases (e.g., distorted or unrecognizable objects), the RealHand dataset requires an additional filtering step. Specifically, we first apply the counting-ready indicator to select hand images with sufficient visual quality, and only then use the counting model to predict the number of fingers, as shown in Figure 2.

## 5 EXPERIMENTS

### 5.1 IMPLEMENTATION DETAILS

**Training of Diffusion Models.** We train a DDPM (Ho et al., 2020) from scratch for the ToyShape dataset and fine-tune latent diffusion models (LDMs) (Rombach et al., 2022), pretrained on the CelebA-HQ dataset (Karras et al., 2018), for the SimObject and RealHand datasets. The time step $T$ is set to $1,000$ with a standard linear noise schedule. Images are resized to 128×128 for DDPMs and 256×256 and LDMs. We train the models using the Adam optimizer (Kingma, 2015) with a learning rate of 0.0001, running for 150k, 300k, and 80k steps on the ToyShape, SimObject, and RealHand datasets, respectively, with an effective batch size of 256.

**Inference of Diffusion Models.** For the ancestral sampling (i.e., DDPM), the sampling steps are set to 1,000. For the ODE-based solvers, we consider both first-order and second-order methods: DPM-Solver-1 (equivalent to DDIM (Song et al., 2021)) as a first-order solver, and DPM-Solver-2 (Lu et al., 2022) as a second-order solver. Each solver is evaluated under three widely-used denoising step configurations: 25, 50, and 100 steps. For the RealHand dataset, the counting model employs a YOLO detector with a confidence threshold of 0.3 and an IoU threshold of 0.1. Unless noted otherwise, the number of generated samples matches the number of training samples in each evaluation, and all experimental results are averaged over three different random seeds.

### 5.2 QUANTIFYING COUNTING HALLUCINATIONS UNDER DIFFERENT DENOISING CONDITIONS

Table 1 presents quantitative results on counting hallucination rate (CHR), non-counting failure rate (NCFR), and total failure rate (TFR) under varying denoising conditions, including solver types, ODE solver orders, and initial noise settings across three datasets: ToyShape, SimObject, and RealHand. Several key observations are as follows.

(1) **Increasing sampling steps reduces counting hallucinations in synthetic datasets but exacerbates them in RealHand**. As shown in Table 1, under commonly used ODE solver settings (25, 50, and 100 steps), increasing the number of sampling steps generally decreases counting hallucination rates in ToyShape and SimObject, but tends to increase it in RealHand, except at 100 steps with DPM-Solver-2. This contrasting behavior suggests that synthetic datasets benefit from finer solver granularity due to their simpler and more structured distributions, whereas real-world datasets such as RealHand exhibit more complex distributions, where additional steps may overfit local inconsistencies and thereby amplify hallucinations.

(2) **Employing higher-order ODE solvers effectively mitigates non-counting failures but exacerbates counting hallucinations in RealHand under the standard inference steps**. As shown in Table 1, comparing DPM-Solver-1 and DPM-Solver-2 on the RealHand dataset at the standard sampling steps (e.g., 25 and 50 steps) reveals a consistent trade-off: while the higher-order solver substantially reduces non-counting failures, it simultaneously leads to a higher counting hallucination rate. A similar trend emerges when increasing the number of sampling steps for both DPM-Solver-1 and DPM-Solver-2. Overall, although using more computation (i.e., "better" sampling conditions) consistently improves overall perceptual quality (i.e., lower non-counting failure rates), it does not enforce factual correctness of finger counts.

(3) **DDPM substantially outperforms ODE solvers in mitigating both counting hallucinations and non-counting failures**. Across all comparisons, ancestral sampling (i.e., DDPM) consistently achieves the lowest CHR, NCFR, and TFR, indicating that DDPM provides a practical lower bound

Table 1: Counting hallucination rates (CHR; the number of counting hallucinations divided by total generated samples) are reported under different solver configurations and initial noise settings across three datasets: ToyShape, SimObject, RealHand. For RealHand, we also report non-counting failure rate (NCFR), which measures failure cases captured by the counting-ready indicator. The total failure rate (TFR) is defined as the sum of CHR and NCFR. *Diffused* and *Normal* refer to the ground-truth initial noise and standard Gaussian noise, respectively.

| Solver Name | Sampling Steps | CHR (ToyShape) | | CHR (SimObject) | | CHR (RealHand) | | NCFR (RealHand) | | TFR (RealHand) | |
|---|---|---|---|---|---|---|---|---|---|---|---|
| | | *Diffused* | *Normal* | *Diffused* | *Normal* | *Diffused* | *Normal* | *Diffused* | *Normal* | *Diffused* | *Normal* |
| DPM-Solver-1 (Song et al., 2021) | 25 | 2.43 | 3.47 | 9.27 | 9.95 | 12.71 | 12.95 | 16.18 | 18.06 | 28.90 | 31.01 |
| | 50 | 1.83 | 2.79 | 8.53 | 9.27 | 13.82 | 13.85 | 11.32 | 12.51 | 25.14 | 26.36 |
| | 100 | 1.56 | 2.32 | 8.04 | 8.90 | 14.29 | 14.55 | 9.54 | 10.63 | 23.83 | 25.19 |
| DPM-Solver-2 (Lu et al., 2022) | 25 | 2.45 | 3.81 | 8.16 | 8.19 | 14.23 | 14.48 | 8.37 | 9.33 | 22.60 | 23.82 |
| | 50 | 1.76 | 2.86 | 7.90 | 7.95 | 16.15 | 15.99 | 6.38 | 7.22 | 22.53 | 23.21 |
| | 100 | 1.41 | 2.24 | 7.89 | 7.89 | 15.06 | 15.43 | 7.82 | 8.94 | 22.88 | 24.38 |
| DDPM (Ho et al., 2020) | 1000 | 0.63 | 0.64 | 4.98 | 4.99 | 10.82 | 10.75 | 2.07 | 2.39 | 12.88 | 13.14 |

for quantifying failure rates of generative models. These results suggest that ancestral sampling serves as an effective strategy for minimizing hallucinations.

(4) **A better initial noise configuration can reduce both counting hallucinations and non-counting failures**. As shown in Table 1, using "diffused" noise (i.e., ground-truth initial noise) tends to results in lower CHR, NCFR, and TFR compared to "normal" noise (i.e., standard Gaussian noise). This effect likely arises because diffused noise better aligns with the initial noise distribution encountered during training, suggesting that a more informed initialization can mitigate not only counting hallucinations but also other non-counting failures.

(5) **Greater morphological complexity in countable objects increases the likelihood of counting hallucinations**. Comparing CHR across ToyShape, SimObject, and RealHand indicates that diffusion models tend to exhibit higher CHR as the morphological complexity of the target objects increases. This suggests that fine-grained details and complex spatial relationships inherent to morphologically complex objects pose additional challenges for diffusion models to maintain accurate object counts, leading to more frequent additions or omissions.

### 5.3 CORRELATION ANALYSIS BETWEEN COUNTING HALLUCINATION RATE, NON-COUNTING FAILURE RATE, AND PERCEPTUAL METRICS

Counting hallucination is a form of factualness hallucination, so it is natural to ask: *Does a better perceptual metric, such as a lower FID, necessarily imply fewer counting hallucinations by reflecting a smaller distribution gap between generated and training data?* The answer is **not necessarily**.

To examine this relationship, we perform correlation analysis between CHR and perceptual metrics, as well as between the NCFR and the same metrics. We focus on two widely used perceptual measures: a distribution-based metric, FID (Heusel et al., 2017), and an instance-based metric, MUSIQ (Ke et al., 2021). For correlation estimation, we adopt three complementary measures: Pearson correlation (Lee Rodgers & Nicewander, 1988) (linear dependence), Spearman rank correlation (Spearman, 1904) (monotonic dependence), and Distance Correlation (Székely et al., 2007) (general nonlinear dependence). The following evidence supports our conclusion.

☞ **The correlations between CHR and perceptual metrics are dataset-dependent and solver-dependent rather than intrinsic**. As shown in the first and third rows of Table 2, the Pearson correlations between CHR and FID vary substantially across datasets (i.e., positive for SimObject, negative for RealHand, and positive again for PTI-Hand), while the corresponding Spearman correlations shift from positive to negative to negative. Distance correlations also decrease from 0.82 (SimObject) to 0.74 (RealHand) to 0.47 (PTI-Hand), suggesting that more complex generative content weakens the association between CHR and FID. Moreover, integrating DDPM results with those from ODE solvers further attenuates these correlations.

☞ **The correlations between NCFR and perceptual metrics are consistently strong and positive**. In contrast to the weak correlations observed between CHR and perceptual metrics, the correlations between NCFR and perceptual metrics is strong and robust across all settings. For example, all

Table 2: Correlation of counting hallucination rate (CHR), non-counting failure rate (NCFR), and total failure rate (TFR) with perceptual metrics, including FID and MUSIQ. Results for PTI-Hand are obtained by prompting a pretrained Stable-Diffusion-3.5-Medium model with ten hand-generation prompts to produce 5050 samples, matching the size of the RealHand dataset. FID for PTI-Hand is computed between RealHand images and the model-generated samples. Here, $p$ denotes the p-value assessing the statistical significance of the correlation $r$ or $\rho$. † means DDPM results integrated with those from various ODE solvers across sampling conditions.

| Metric | CHR vs. FID | | | | | | NCFR vs. FID | | | | CHR vs. MUSIQ | | | | NCFR vs. MUSIQ | | | |
|---|---|---|---|---|---|---|---|---|---|---|---|---|---|---|---|---|---|---|
| | SimObject | | RealHand | | PTI-Hand | | RealHand | | PTI-Hand | | RealHand | | PTI-Hand | | RealHand | | PTI-Hand | |
| | $r/\rho/r_d$ | $p$ | $r/\rho/r_d$ | $p$ | $r/\rho/r_d$ | $p$ | $r/\rho/r_d$ | $p$ | $r/\rho/r_d$ | $p$ | $r/\rho/r_d$ | $p$ | $r/\rho/r_d$ | $p$ | $r/\rho/r_d$ | $p$ | $r/\rho/r_d$ | $p$ |
| Pearson ($r$) | 0.87 | 0.01 | -0.97 | <0.01 | 0.20 | 0.69 | 0.99 | <0.01 | 0.88 | <0.01 | 0.93 | <0.01 | -0.02 | 0.95 | -0.97 | <0.01 | -0.99 | <0.01 |
| Pearson† ($r$) | 0.49 | 0.14 | -0.53 | 0.13 | – | – | 0.96 | <0.01 | – | – | 0.46 | 0.20 | – | – | -0.98 | <0.01 | – | – |
| Spearman ($\rho$) | 0.84 | <0.01 | -0.97 | <0.01 | -0.14 | 0.78 | 0.99 | <0.01 | 0.92 | <0.01 | 0.80 | 0.01 | 0.42 | 0.33 | -0.88 | <0.01 | -0.46 | 0.29 |
| Spearman† ($\rho$) | 0.75 | 0.01 | -0.51 | 0.15 | – | – | 0.95 | <0.01 | – | – | 0.40 | 0.28 | – | – | -0.91 | <0.01 | – | – |
| Distance Correlation ($r_d$) | 0.89 | – | 0.97 | – | 0.47 | – | 0.99 | – | 0.91 | – | 0.93 | – | 0.42 | – | 0.97 | – | 0.99 | – |
| Distance Correlation† ($r_d$) | 0.82 | – | 0.74 | – | – | – | 0.95 | – | – | – | 0.71 | – | – | – | 0.97 | – | – | – |

(a) O1: FR vs. Steps    (b) O1: Quality vs. Steps    (c) O2: FR vs. Steps    (d) O2: Quality vs. Steps

(e) 25 steps: FR vs. Order    (f) 25 steps: Quality vs. Order

Figure 3: Comparison of trend consistency between DM-RealHand (ours) and PTI-Hand (pretrained Stable-Diffusion-3.5-Medium). O1 and O2 denote first- and second-order ODE solvers (DPM-solver-1 / DPM-solver-2 for DM-RealHand; Euler / Heun for PTI-Hand). FR metrics include CHR and NCFR; quality metrics include FID and MUSIQ. Across variations in sampling steps and solver order, both models exhibit highly consistent trends.

correlation coefficients are over 0.88 in NCFR vs. FID, with very small p-values, indicating high statistical significance.

## 5.4 GENERALIZATION OF REALHAND FINDINGS TO PRETRAINED TEXT-TO-IMAGE MODELS

In this section, we evaluate counting hallucinations in a large-scale pretrained text-to-image model. Specifically, we assess Stable-Diffusion-3.5-Medium (Esser et al., 2024) using the counting-ready indicator and counting model introduced in Sec. 4.2.3. More details are provided in Appendix A.8. As shown in Figure 3, the way solver type and sampling steps affect counting hallucinations in large pretrained diffusion models aligns closely with the trends revealed by our controlled RealHand experiments. This consistency highlights the validity of the findings presented in Sec. 5.2 and underscores the broader applicability of our evaluation protocol. Correlation results in Table 2 further support this conclusion. In specific, weak correlations between CHR and perceptual metrics and strong correlations between NCFR and perceptual metrics are also observed. Together, these results demonstrate the robustness and generality of our findings and show that the proposed evaluation protocol transfers reliably to pretrained text-to-image models.

## 5.5 HOW TO REDUCE COUNTING HALLUCINATIONS?

The lower counting hallucination rates observed for simpler countable objects (see Observation 5 in Sec. 5.2) suggest that diffusion models more reliably capture counting-based factuality when the underlying visual structure exhibits low morphological complexity (e.g., masked objects). This

Figure 4: Diagram of the proposed joint-diffusion model (JDM). The model performs channel-wise concatenation of the latent representations of hand RGB images and their corresponding segmentation masks. This joint learning paradigm encourages anatomically plausible hand poses and promotes correct finger topology during generation, resulting in reduced CHR and NCFR.

Table 3: Comparison of failure rates between LDM and our proposed JDM on the RealHand dataset across different ODE solvers and sampling steps. Lower values indicate better performance. For JDM, although the model jointly generates hand RGB images and segmentation masks, FID is computed solely on the RGB outputs to ensure comparability with baselines.

| | Counting Hallucination Rate (CHR) ↓ | | | | | | Non-counting Failure Rate (NCFR) ↓ | | | | | | Fréchet Inception Distance (FID) ↓ | | | | | |
|---|---|---|---|---|---|---|---|---|---|---|---|---|---|---|---|---|---|---|
| Solver Name | DPM-Solver-1 | | | DPM-Solver-2 | | | DPM-Solver-1 | | | DPM-Solver-2 | | | DPM-Solver-1 | | | DPM-Solver-2 | | |
| Sampling Steps | 25 | 50 | 100 | 25 | 50 | 100 | 25 | 50 | 100 | 25 | 50 | 100 | 25 | 50 | 100 | 25 | 50 | 100 |
| LDM | 12.95 | 13.85 | 14.55 | 14.48 | 15.99 | 15.43 | 18.06 | 12.51 | 10.63 | 9.33 | 7.22 | 8.94 | **29.86** | **26.43** | **25.21** | **24.48** | **23.42** | **24.54** |
| JDM (ours) | **11.41** | **10.67** | **10.51** | **10.12** | **9.66** | **10.07** | **2.94** | **2.42** | **2.29** | **3.51** | **4.63** | **2.26** | 37.69 | 35.51 | 34.68 | 34.63 | 32.07 | 33.89 |

finding raises an important question: if diffusion models were to denoise general visual representations together with explicit, simple structural constraints that are easier to model, such as object masks that encode basic topology and counts, within a shared latent space, could such a joint learning paradigm mitigate counting hallucinations? We refer to such models that jointly learn both general visual representations and structural constraints during denoising as **joint-diffusion models** (JDM).

Specifically, we perform a channel-wise concatenation of the latent representation of hand mask with that of the corresponding RGB image, as shown in Fig. 4. Then we finetune the UNet pretrained on CelebA-HQ (Karras et al., 2018) (following the RealHand configuration described in Sec. 5.1) using the RealHand images and their associated masks, extracted using SAM-2 (Ravi et al., 2024). To accommodate both modalities, the input and output channels of the denoising UNet are expanded to match the combined latent dimensionality of the image and mask latents (i.e., 3+3=6 channels).

As shown in Table 3, JDM consistently achieves lower failure rates than the vanilla LDM, including CHR and NCFR, across diverse sampling conditions, highlighting its effectiveness in accurately generating structurally intricate objects that are prone to counting ambiguities. On the other hand, JDM yields higher FID. This degradation likely stems from the increased denoising burden introduced by jointly modeling two modalities under a small-scale training regime, which forces the model to trade off perceptual quality for improved structural realism. We also observe promising reductions in counting hallucinations when applying JDM to large-scale pretrained diffusion models under carefully controlled finetuning settings. More details, results and discussion can be found in Appendix A.9.

## 6 CONCLUSION

In this work, we take the first step toward systematically quantifying hallucinations in diffusion models and offer new insights into improving factual generation in this field. Focusing on counting hallucinations, we introduce **CountHalluSet** and a practical and reproducible methodology for measuring counting hallucinations in practice. Through comprehensive and controlled quantification experiments, we show that commonly adopted numerical strategies often fail to mitigate counting hallucinations in generating human hands. Moreover, our correlation analyses demonstrate that widely used perceptual metrics (e.g., FID and MUSIQ) are unable to consistently reflect the severity of counting hallucinations. Experiments on large-scale pretrained text-to-image models further corroborate both the generality and the validity of our findings.

## ETHICS STATEMENT

This study complies with the ethical standards of ICLR. No human or animal participants were involved in the research. The datasets introduced in this work, namely CountHalluSet (including ToyShape, SimObject, and RealHand), are either synthetically created or collected from open repositories (e.g., 11k Hands, Kaggle, Roboflow) under their respective terms of use. All data were carefully processed to avoid including any personal or sensitive information. The experiments presented in this paper do not pose risks to privacy, safety, or security.

## REPRODUCIBILITY STATEMENT

We have made every effort to ensure that the results reported in this paper are reproducible. The CountHalluSet dataset, including ToyShape, SimObject, and RealHand, has been described in detail with counting rules and construction protocols. The experimental setup, including model architectures, training schedules, sampling conditions (solver type, solver order, sampling steps, and noise settings), and evaluation protocols, is fully detailed in the paper. We will release the datasets, code, and pre-trained counting models upon publication to support transparent replication and verification. This ensures that other researchers can reproduce our experiments and build upon our work.

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

# A APPENDIX

## A.1 POOF OF THE BOUND OF TOTAL ACCUMULATED DISTRIBUTIONAL ERROR IN ANCESTRAL SAMPLING

For the joint distributions of the ideal reverse process $q(\boldsymbol{x}_{T:0})$ and the true reverse process $p_\theta(\boldsymbol{x}_{T:0})$, their KL divergence is defined as:

$$\mathcal{D}_{\text{KL}}(q(\boldsymbol{x}_{T:0})\|p_\theta(\boldsymbol{x}_{T:0})) = \mathbb{E}_{q(\boldsymbol{x}_{T:0})}\left[log\frac{q(\boldsymbol{x}_{T:0})}{p_\theta(\boldsymbol{x}_{T:0})}\right]. \tag{A.1}$$

For both reverse processes $q(\boldsymbol{x}_{T:0})$ and $p_\theta(\boldsymbol{x}_{T:0})$, we can decompose them as follows, leveraging the Markov property:

$$q(\boldsymbol{x}_{T:0}) = q(\boldsymbol{x}_T)\prod_{t=1}^{T} q(\boldsymbol{x}_{t-1} \mid \boldsymbol{x}_t), \tag{A.2}$$

$$p_\theta(\boldsymbol{x}_{T:0}) = p_\theta(\boldsymbol{x}_T)\prod_{t=1}^{T} p_\theta(\boldsymbol{x}_{t-1} \mid \boldsymbol{x}_t). \tag{A.3}$$

By applying the chain rule to decompose the KL divergence, we have:

$$\mathcal{D}_{\text{KL}}(q(\boldsymbol{x}_{T:0})\|p_\theta(\boldsymbol{x}_{T:0})) = \mathbb{E}_{q(\boldsymbol{x}_{T:0})}\left[log\frac{q(\boldsymbol{x}_T)\prod_{s=1}^{T} q(\boldsymbol{x}_{s-1} \mid \boldsymbol{x}_s)}{p_\theta(\boldsymbol{x}_T)\prod_{s=1}^{T} p_\theta(\boldsymbol{x}_{s-1} \mid \boldsymbol{x}_s)}\right]$$

$$= \mathbb{E}_{q(\boldsymbol{x}_{T:0})}\left[log\frac{q(\boldsymbol{x}_T)}{p_\theta(\boldsymbol{x}_T)} + \sum_{t=1}^{T} log\frac{q(\boldsymbol{x}_{t-1} \mid \boldsymbol{x}_t)}{p_\theta(\boldsymbol{x}_{t-1} \mid \boldsymbol{x}_t)}\right]$$

$$= \mathbb{E}_{q(\boldsymbol{x}_T)}\left[log\frac{q(\boldsymbol{x}_T)}{p_\theta(\boldsymbol{x}_T)}\right] + \sum_{t=1}^{T} \mathbb{E}_{q(\boldsymbol{x}_t)}\left[\mathbb{E}_{q(\boldsymbol{x}_{t-1}|\boldsymbol{x}_t)}\left[log\frac{q(\boldsymbol{x}_{t-1} \mid \boldsymbol{x}_t)}{p_\theta(\boldsymbol{x}_{t-1} \mid \boldsymbol{x}_t)}\Big|\boldsymbol{x}_t\right]\right]$$

$$= \mathcal{D}_{\text{KL}}(q(\boldsymbol{x}_T\|p_\theta(\boldsymbol{x}_T)) + \sum_{t=1}^{T} \mathbb{E}_{q(\boldsymbol{x}_t)}[\mathcal{D}_{\text{KL}}(q(\boldsymbol{x}_{t-1} \mid \boldsymbol{x}_t)\|p_\theta(\boldsymbol{x}_{t-1} \mid \boldsymbol{x}_t))]. \tag{A.4}$$

Then, according to the Data Processing Inequality (Cover & Thomas, 2012), marginalizing (or discarding some random variables) will not increase the KL divergence:

$$\mathcal{D}_{\text{KL}}(q(\boldsymbol{x}_0)\|p_\theta(\boldsymbol{x}_0)) \leq \mathcal{D}_{\text{KL}}(q(\boldsymbol{x}_{T:0})\|p_\theta(\boldsymbol{x}_{T:0})). \tag{A.5}$$

**The bound of total accumulated distributional error** can be obtained by combining Eq. (A.4) and Eq. (A.5):

$$\mathcal{D}_{\text{KL}}(q(\boldsymbol{x}_0)\|p_\theta(\boldsymbol{x}_0)) \leq \underbrace{\mathcal{D}_{\text{KL}}(q(\boldsymbol{x}_T)\|p_\theta(\boldsymbol{x}_T))}_{\text{diffused-prior gap}} + \underbrace{\sum_{t=1}^{T} \mathbb{E}_{q(\boldsymbol{x}_t)}[\mathcal{D}_{\text{KL}}(q(\boldsymbol{x}_{t-1} \mid \boldsymbol{x}_t)\|p_\theta(\boldsymbol{x}_{t-1} \mid \boldsymbol{x}_t))]}_{\text{accumulated transition errors caused by the model prediction error}}, \tag{A.6}$$

where equality (=) would strictly hold only in the ideal case where $q(\boldsymbol{x}_T) = p_\theta(\boldsymbol{x}_T)$ and $p_\theta(\boldsymbol{x}_{t-1} \mid \boldsymbol{x}_t) = q(\boldsymbol{x}_{t-1} \mid \boldsymbol{x}_t)$ for all $t$ and $\boldsymbol{x}_t$, which means the diffusion model can accurately recover the underlying data distribution $q(\boldsymbol{x}_0)$. This upper bound for the overall KL divergence ($\mathcal{D}_{\text{KL}}(q(\boldsymbol{x}_0)\|p_\theta(\boldsymbol{x}_0))$) between true data and model-generated data is identical to the variational bound derived in DDPM (Ho et al., 2020) when data entropy $H(\boldsymbol{x}_0)$ is excluded, thereby suggesting that optimizing DDPM's variational loss directly tightens this explicit upper bound on the discrepancy between trued data and model-generated data.

**Diffused-prior gap.** The first source of errors arises from the mismatch between the distribution of the ground-truth diffused data $q_T(\boldsymbol{x}_T) = q_{0:T}(\boldsymbol{x}_T|\boldsymbol{x}_0)$ and the prior distribution used for sampling $p_\theta(\boldsymbol{x}_T)$. This mismatch stems from the fact that the number of diffusion steps $T$ is finite in practice. The discrepancy can be formally quantified using a general distributional metric:

$$\mathcal{D}_{\text{KL}}(q_T(\boldsymbol{x}_T)\|p_\theta(\boldsymbol{x}_T)), \quad p_\theta(\boldsymbol{x}_T) = \mathcal{N}(\boldsymbol{0}, \boldsymbol{I}), \tag{A.7}$$

where $\mathcal{D}_{\mathrm{KL}}$ denotes the Kullback–Leibler (KL) divergence. Alternatively, other appropriate metrics such as FID or the Wasserstein distance can be employed to evaluate this distributional discrepancy. The diffused-prior gap, present from the outset, induces a deviation that persists throughout the denoising process, ultimately resulting in a sustained output error.

**Noise model prediction error.** The second source of errors stems from the noise prediction model. Because the model's estimate $\epsilon_\theta(\boldsymbol{x}_t, t)$ only approximates the true noise $\epsilon$, as guaranteed in principle by the *Universal Approximation Theorem* (Hornik et al., 1989; Nielsen, 2015; Yarotsky, 2022). Let $\Delta\epsilon_t(\boldsymbol{x}_t) = \epsilon - \epsilon_\theta(\boldsymbol{x}_t, t)$ represent this prediction error. Based on Eq. (5), any discrepancy in $\epsilon_\theta$ induces a corresponding deviation in the predicted mean at step $t$:

$$\Delta\boldsymbol{\mu}_t(\boldsymbol{x}_t) = -\frac{\beta_t}{\sqrt{1 - \beta_t}} \frac{1}{\sqrt{1 - \bar{\alpha}_t}} \Delta\epsilon_t(\boldsymbol{x}_t). \tag{A.8}$$

Since each reverse step uses the mean from the previous iteration, this bias accumulates throughout the denoising trajectory, ultimately contributing to the overall prediction error.

## A.2 Poof of the Total Accumulated Trajectory Error in ODE-Based Sampling

Based on Eq. (7), we take a one-step numerical solver approximates the solution as example. Let $\boldsymbol{x}^*(\tau)$ denote the ideal trajectory that satisfies this ODE with the true score function $h^*(\boldsymbol{x}(\tau), \tau)$ and starts from a ground-truth initial condition $\boldsymbol{x}^*(T)$ drawn from the true diffused distribution $q_T(\boldsymbol{x}_T)$. Similarly, $\hat{\boldsymbol{x}}_k$ represent the numerical trajectory computed by a numerical ODE solver at discrete time steps $\tau_k = T - k\Delta t$ for $k = 0, 1, \ldots, N$ and $\hat{h}$ is the estimated model function. For a specific numerical trajectory from $\hat{\boldsymbol{x}}_0 - \hat{\boldsymbol{x}}_N$, we want to calculate its total accumulated trajectory error. Let $e_k = \hat{\boldsymbol{x}}_k - \boldsymbol{x}^*(\tau_k)$ be the trajectory error at step $k$. Consider the transition from $\tau_k$ to $\tau_{k+1} = \tau_k - \Delta t$.

**Ideal path evolution.** Using the variation-of-constants formula for the exact solution over one step (from $\tau_k$ down to $\tau_{k+1}$):

$$\boldsymbol{x}^*(\tau_{k+1}) = \mathcal{G}(\tau_{k+1}, \tau_k)\boldsymbol{x}^*(\tau_k) + \int_{\tau_k}^{\tau_{k+1}} \mathcal{G}(\tau_{k+1}, u)h^*(\boldsymbol{x}^*(u), u)\mathrm{d}u, \tag{A.9}$$

where $\mathcal{G}(t_2, t_1) = e^{\int_{t_1}^{t_2} f(v)\mathrm{d}v}$ is the state transition operator for the linear part $\frac{\mathrm{d}\boldsymbol{z}}{\mathrm{d}\tau} = f(\tau)\boldsymbol{z}$. Note that the integration limits are backward in time.

**Numerical path evolution.** A one-step numerical solver approximates the solution. This numerical step deviates from the exact evolution governed by $\hat{h}$ starting from $\hat{\boldsymbol{x}}_k$. Let $\hat{\boldsymbol{x}}_{\mathrm{exact}}(\tau_{k+1} \mid \hat{\boldsymbol{x}}_k, \hat{h})$ be the exact solution at $\tau_{k+1}$ of the ODE using $\hat{h}$, starting from $\hat{\boldsymbol{x}}_k$ at $\tau_k$:

$$\hat{\boldsymbol{x}}_{\mathrm{exact}}(\tau_{k+1} \mid \hat{\boldsymbol{x}}_k, \hat{h}) = \mathcal{G}(\tau_{k+1}, \tau_k)\hat{\boldsymbol{x}}_k + \int_{\tau_k}^{\tau_{k+1}} \mathcal{G}(\tau_{k+1}, u)\hat{h}(\hat{\boldsymbol{x}}_{\mathrm{exact}}(u \mid \hat{\boldsymbol{x}}_k, \hat{h}), u)\mathrm{d}u. \tag{A.10}$$

The numerical method introduces a local truncation error $\delta_k$:

$$\hat{\boldsymbol{x}}_{k+1} = \hat{\boldsymbol{x}}_{\mathrm{exact}}(\tau_{k+1} \mid \hat{\boldsymbol{x}}_k, \hat{h}) + \delta_k. \tag{A.11}$$

So, we have:

$$\hat{\boldsymbol{x}}_{k+1} = \mathcal{G}(\tau_{k+1}, \tau_k)\hat{\boldsymbol{x}}_k + \int_{\tau_k}^{\tau_{k+1}} \mathcal{G}(\tau_{k+1}, u)\hat{h}(\hat{\boldsymbol{x}}_{\mathrm{exact}}(u \mid \hat{\boldsymbol{x}}_k, \hat{h}), u)\mathrm{d}u + \delta_k. \tag{A.12}$$

**Error recurrence.** Subtract the ideal evolution Eq. (A.9) from the numerical evolution Eq. (A.12):

$$e_{k+1} = \hat{\boldsymbol{x}}_{k+1} - \boldsymbol{x}^*(\tau_{k+1})$$

$$= \mathcal{G}(\tau_{k+1}, \tau_k)(\hat{\boldsymbol{x}}_k - \boldsymbol{x}^*(\tau_k)) + \int_{\tau_k}^{\tau_{k+1}} \mathcal{G}(\tau_{k+1}, u)\left[\hat{h}(\hat{\boldsymbol{x}}_{\mathrm{exact}}(u \mid \hat{\boldsymbol{x}}_k, \hat{h})) - h^*(\boldsymbol{x}^*(u), u)\right]\mathrm{d}u + \delta_k. \tag{A.13}$$

**Accumulated error formulation.** We can unroll this recurrence relation from $k = 0$ to $N - 1$. Let $\mathcal{G}(\tau_j, \tau_k) = \mathcal{G}(\tau_j, \tau_{j-1}), \ldots, \mathcal{G}(\tau_{k+1}, \tau_k)$ for $j > k$, and $\mathcal{G}(\tau_k, \tau_k) = \boldsymbol{I}$. Applying the recurrence

repeatedly:

$$e_N = \mathcal{G}(\tau_N, \tau_0)e_0 + \sum_{k=0}^{N-1} \mathcal{G}(\tau_N, \tau_{k+1})$$
$$\cdot \left( \int_{\tau_k}^{\tau_{k+1}} \mathcal{G}(\tau_{k+1}, u) \left[ \hat{h}(\hat{\boldsymbol{x}}_{\text{exact}}(u \mid \hat{\boldsymbol{x}}_k, \hat{h})) - h^*(\boldsymbol{x}^*(u), u) \right] \mathrm{d}u + \delta_k \right). \quad (A.14)$$

Substituting $\tau_N = 0$, $\tau_0 = T$, $e_N = \hat{\boldsymbol{x}_N} - \boldsymbol{x}^*(0)$, and $e_0 = \hat{\boldsymbol{x}}_0 - \boldsymbol{x}^*(T)$:

$$\hat{\boldsymbol{x}}_N - \boldsymbol{x}^*(0) = \underbrace{\mathcal{G}(0, T)(\hat{\boldsymbol{x}}_0 - \boldsymbol{x}^*(T))}_{\text{propagated initial error}} + \sum_{k=0}^{N-1} \mathcal{G}(0, \tau_{k+1})$$

$$\cdot \left( \underbrace{\int_{\tau_k}^{\tau_{k+1}} \mathcal{G}(\tau_{k+1}, u) \left[ \hat{h}(\hat{\boldsymbol{x}}_{\text{exact}}(u \mid \hat{\boldsymbol{x}}_k, \hat{h}), u) - h^*(\boldsymbol{x}^*(u), u) \right] \mathrm{d}u}_{\text{single step model prediction error contribution}} + \underbrace{\delta_k}_{\substack{\text{single step truncation error}}} \right).$$
$$(A.15)$$

where the propagated initial error term represents the deterministic impact on the final path error of a specific initial error instance, whose statistical origin lies in the distributional Diffused-Prior Gap.

## A.3 THE DISCREPANCY BETWEEN DDPM AND DDIM IN THE FORWARD AND REVERSE PROCESS

As shown in Eq. (2), the forward process in DDPM (Ho et al., 2020) can be formulated as:

$$\boldsymbol{x}_t = \sqrt{\bar{\alpha}_t}\boldsymbol{x}_0 + (1 - \bar{\alpha}_t)\boldsymbol{\epsilon}, \quad \boldsymbol{\epsilon} \sim \mathcal{N}(\boldsymbol{0}, \boldsymbol{I}). \quad (A.16)$$

And the reverse process is defined as follows:

$$\boldsymbol{x}_{t-1}^{\text{DDPM}} = \frac{1}{\sqrt{\alpha_t}}\boldsymbol{x}_t - \frac{1 - \alpha_t}{\sqrt{\alpha_t}\sqrt{1 - \bar{\alpha}_t}}\boldsymbol{\epsilon}_\theta(\boldsymbol{x}_t, t) + \sqrt{\beta_t}\boldsymbol{\epsilon}. \quad (A.17)$$

For DDIM (Song et al., 2021), the forward process is identical to that in Eq. (A.16), while its deterministic reverse process (with $\eta = 0$) is defined as follows:

$$\boldsymbol{x}_{t-1}^{\text{DDIM}} = \sqrt{\frac{\bar{\alpha}_{t-1}}{\bar{\alpha}_t}}(\boldsymbol{x}_t - \sqrt{1 - \bar{\alpha}_t}\boldsymbol{\epsilon}_\theta(\boldsymbol{x}_t, t)) + \sqrt{1 - \bar{\alpha}_{t-1}}\boldsymbol{\epsilon}_\theta(\boldsymbol{x}_t, t)$$

$$= \frac{1}{\sqrt{\alpha_t}}\boldsymbol{x}_t - \left( \frac{\sqrt{1 - \bar{\alpha}_t}}{\sqrt{\alpha_t}} - \sqrt{1 - \bar{\alpha}_{t-1}} \right)\boldsymbol{\epsilon}_\theta(\boldsymbol{x}_t, t). \quad (A.18)$$

Although DDPM and DDIM share the same forward (diffusion) process, DDIM does not simply strip the stochastic term from DDPM's reverse (denoising) process. This is because the two methods employ different coefficient formulations for the predicted noise term.

## A.4 DATASET EXAMPLES AND DETAILS

Figure A1 presents representative examples from the three datasets used in this study. As shown in Table A1, each dataset exhibits a balanced distribution across its respective categories. Table A2 further demonstrates that the distribution of images with different object counts is comparable, indicating the absence of object-count bias across datasets. The datasets appear in the same order in both tables, and the category order in Table A1 is defined as follows:

- **ToyShape**: triangle, square, pentagon;
- **SimObject**: mug, apple, clock;
- **RealHand**: finger.

Table A1: Number of objects per category for each dataset.

|  | ToyShape | SimObject | RealHand |
|---|---|---|---|
| Category 1 | 20,027 | 19,916 | 25,250 |
| Category 2 | 19,902 | 20,101 | - |
| Category 3 | 19,978 | 19,974 | - |

Table A2: Number of images by object count(s) per dataset.

|  | ToyShape | SimObject | RealHand |
|---|---|---|---|
| 1 object | 10,003 | 10,022 | - |
| 2 objects | 10,087 | 9,965 | - |
| 3 objects | 9,910 | 10,013 | - |
| 5 objects | - | - | 25,250 |
| Total | 30,000 | 30,000 | 25,250 |

## A.5 EXAMPLES OF COUNTING HALLUCINATIONS

We provide some representative examples of counting hallucinations corresponding to three datasets, as shown in Fig. A2. For the ToyShape and SimObject datasets, the counting hallucinations contain more than two instances of at least one category or do not contain any instances but with normal background, as shown in Fig. A2(a) and Fig. A2(b). In the Realhand dataset, counting-hallucinated samples are characterized by incorrect finger counts (e.g., 4 or 6 fingers), as shown in Fig. A2(c). Notably, the last two images of the first row in Fig. A2(c) appear to display five fingers; however, one finger is partially incomplete and visibly artifacted, effectively resulting in four valid fingers. Consequently, these samples are classified as counting hallucinations.

## A.6 EXAMPLES OF YOLO PREDICTIONS OF FINGERTIPS

Figure A3 presents representative examples of YOLO predictions on the RealHand dataset. The YOLO detector accurately identifies fingertips across diverse hand poses and orientations, including both dorsal and palmar views. However, it may still fail to recognize anatomically implausible configurations, such as hands with duplicated thumbs (e.g., the penultimate sub-figure of the first row and the last sub-figure of the last row in Fig. A3(a)) or hands with five middle fingers but without a thumb (e.g., the last sub-figure of the first row and the penultimate sub-figure of the last row in Fig. A3(a)). These abcounting-valid samples are still detected as five-finger hands and therefore remain indistinguishable from anatomically valid ones in terms of finger counts.

Figure A3(b) shows several counting-hallucinated samples where YOLO predicts incorrect fingertip counts (e.g., 3, 4, 6, or 7 fingertips). Figure A3(c) further illustrates several rare edge cases (<1%) in which the detector fails to identify fingertips with unclear boundaries, cracked fingers, or floating artifacts. Such rare cases can be partly mitigated by tuning YOLO's inference hyperparameters (e.g., IoU and confidence thresholds); however, doing so may compromise detection performance on the majority of counting-valid samples. Since a consistent detection setup is applied across all evaluations on the RealHand dataset, the presence of these rare cases does not affect the validity of our overall conclusions, even though addressing them remains a challenging and low-reward effort.

## A.7 EXAMPLES OF THREE CLASSES OF GENERATED HAND IMAGES

Figure A4 shows representative examples of three classes of generated hand images: counting-valid, counting-hallucinated, and non-counting failure samples. As illustrated in Fig. 2, the evaluation procedure of the RealHand dataset first applies the counting-ready indicator to separate counting-ready samples (i.e., counting-valid and counting-hallucinated) from images unsuitable for counting. The latter typically exhibit severe visual artifacts, such as distorted, cracked, partially visible, or floating fingers. Counting-ready samples are then further classified as counting-valid or counting-hallucinated based on whether the detected number of valid fingertips equals five.

## A.8 Extended Details of PTI-Hand experiments

In the PTI-Hand experiments, we adopt 10 different hand-generation prompts to prompt Stable-Diffusion-3.5-Medium. The prompts are show as follows.

1. `A single human hand, front view, all fingers in frame and visible.`

2. `A complete human hand, seen from the front, showing all fingers clearly, minimal background.`

3. `An isolated single hand, viewed from directly above, showing all fingers, simple studio background.`

4. `Overhead shot of a person's hand, entire hand in frame, all fingers visible, plain background.`

5. `One hand, top view, five fingers visible, simple background.`

6. `A single hand, front or back view, flat against a surface, all fingers in frame and fully visible.`

7. `A single human hand, top-down view, all fingers visible, resting on a plain white surface.`

8. `A single human hand seen from above, fingers spread naturally, minimal background.`

9. `A human hand, top view, palm facing downward, all fingers clearly in frame, neutral background.`

10. `A single human hand captured from overhead, fingers extended, plain background with soft lighting.`

The definition of hallucinations assumes that the model generates patterns that were never observed during training. However, this assumption is difficult to verify for large-scale pretrained diffusion models, where the full training distribution is inaccessible. To make the quantification of counting hallucinations in such cases feasible, we adopt a conservative criterion: we treat any generated hand image containing six or more fingers as a counting-hallucinated sample. This assumption is well justified, while real images may depict fewer than five visible fingers due to occlusion, articulation, or viewpoint, it is highly unlikely for real data to contain hands with six or more fingers. Thus, finger counts $\geq 6$ provide a reliable and empirically verifiable signal of counting hallucination. Generated examples can be found in Fig. A5.

## A.9 Extended Details, Results, and Discussion of the Joint Diffusion Model

### A.9.1 Implementation details in Table 3

In the JDM experiments reported in Table 3, we fine-tune the denoising UNet for 120k steps, compared to 80k steps for the vanilla LDM. This increased training budget is necessary because extending the latent space from three to six channels in JDM introduces a substantial mismatch from the pretrained latent distribution of LDM. As a result, additional optimization is required to close this distributional gap and allow JDM's modeling capacity to fully materialize.

### A.9.2 Fine-Tuning Pretrained Text-to-Image Models with Joint-Diffusion Training

Table A3: Comparison of failure rates and image quality between finetuning SD-1.5 and finetuning SD-1.5 with joint-diffusion training paradigm on RealHand. The first row (baseline) reports results from finetuning Stable-Diffusion-1.5 on RealHand data. The second row shows results obtained by freezing all intermediate layers of the baseline model and finetuning only the input and output layers with extended channel dimensions. Lower values indicate better performance.

| Solver Name | Counting Hallucination Rate (CHR) ↓ | | | | | | Non-counting Failure Rate (NCFR) ↓ | | | | | | Fréchet Inception Distance (FID) ↓ | | | | | |
| --- | --- | --- | --- | --- | --- | --- | --- | --- | --- | --- | --- | --- | --- | --- | --- | --- | --- | --- |
| | DPM-Solver-1 | | | DPM-Solver-2 | | | DPM-Solver-1 | | | DPM-Solver-2 | | | DPM-Solver-1 | | | DPM-Solver-2 | | |
| Sampling Steps | 25 | 50 | 100 | 25 | 50 | 100 | 25 | 50 | 100 | 25 | 50 | 100 | 25 | 50 | 100 | 25 | 50 | 100 |
| SD-1.5 | 0.3362 | 0.3404 | **0.3630** | 0.3396 | 0.3376 | **0.3362** | 0.0059 | 0.0032 | 0.0020 | 0.0050 | 0.0026 | 0.0018 | 61.51 | 57.71 | 57.70 | 58.53 | 57.57 | 56.69 |
| SD-1.5 with joint-diffusion finetuning | **0.2838** | **0.3222** | 0.4145 | **0.3053** | **0.3277** | 0.3558 | 0.0947 | 0.0511 | 0.0487 | 0.0549 | 0.0362 | 0.0327 | 70.65 | 79.22 | 102.39 | 70.80 | 72.84 | 83.07 |

Importantly, JDM should be viewed as **a novel training paradigm** for improving factual generation in diffusion models rather than as a superior inference strategy. On the one hand, JDM only extends

the input and output channels of the UNet while keeping all sizes of intermediate layers unchanged, resulting in **introduces a negligible inference-time overhead compared to vanilla LDM**. On the other hand, extending a pretrained text-to-image model with additional mask channels **disrupts the manifold geometry of the pretrained latent space and underlines the original denoising distribution**, making direct finetuning such model nontrivial.

Despite this challenge, we still observe promising reductions in counting hallucinations when applying JDM to large-scale pretrained diffusion models under carefully controlled finetuning conditions. In practice, we find that jointly aligning the extended latent space in JDM with the highly rigid latent geometry of large pretrained models is challenging under the limited scale of RealHand. To strike a practical balance, we adopt a two-stage finetuning strategy: we first finetune a Stable-Diffusion-1.5 model on RealHand, then freeze all intermediate UNet layers and finetune only the input and output layers with extended channels. As shown in Table A3, even under this highly constrained finetuning setup, the joint-diffusion paradigm consistently yields lower counting hallucination rates at 25 and 50 sampling steps across both solvers.

Overall, the evidence presented in this work suggests that JDM is a promising direction for enforcing factual consistency in image generation. Lastly, we believe that **exploring its behavior and performance under true large-scale training remains an important and compelling avenue for future research**.

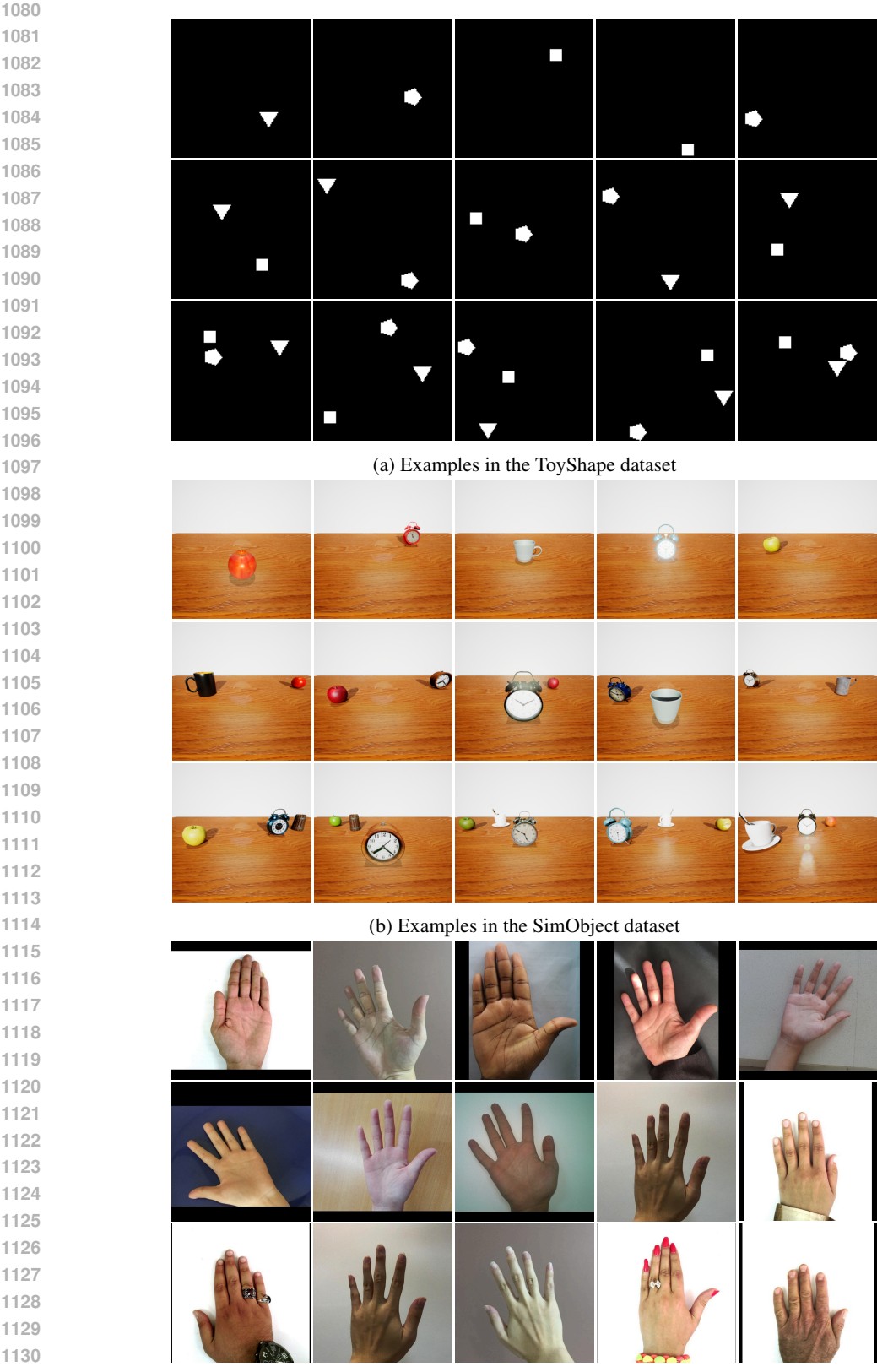

(a) Examples in the ToyShape dataset

(b) Examples in the SimObject dataset

(c) Examples in the RealHand dataset

Figure A1: Examples of datasets in the CountHalluSet suite: ToyShape, SimObject, and RealHand.

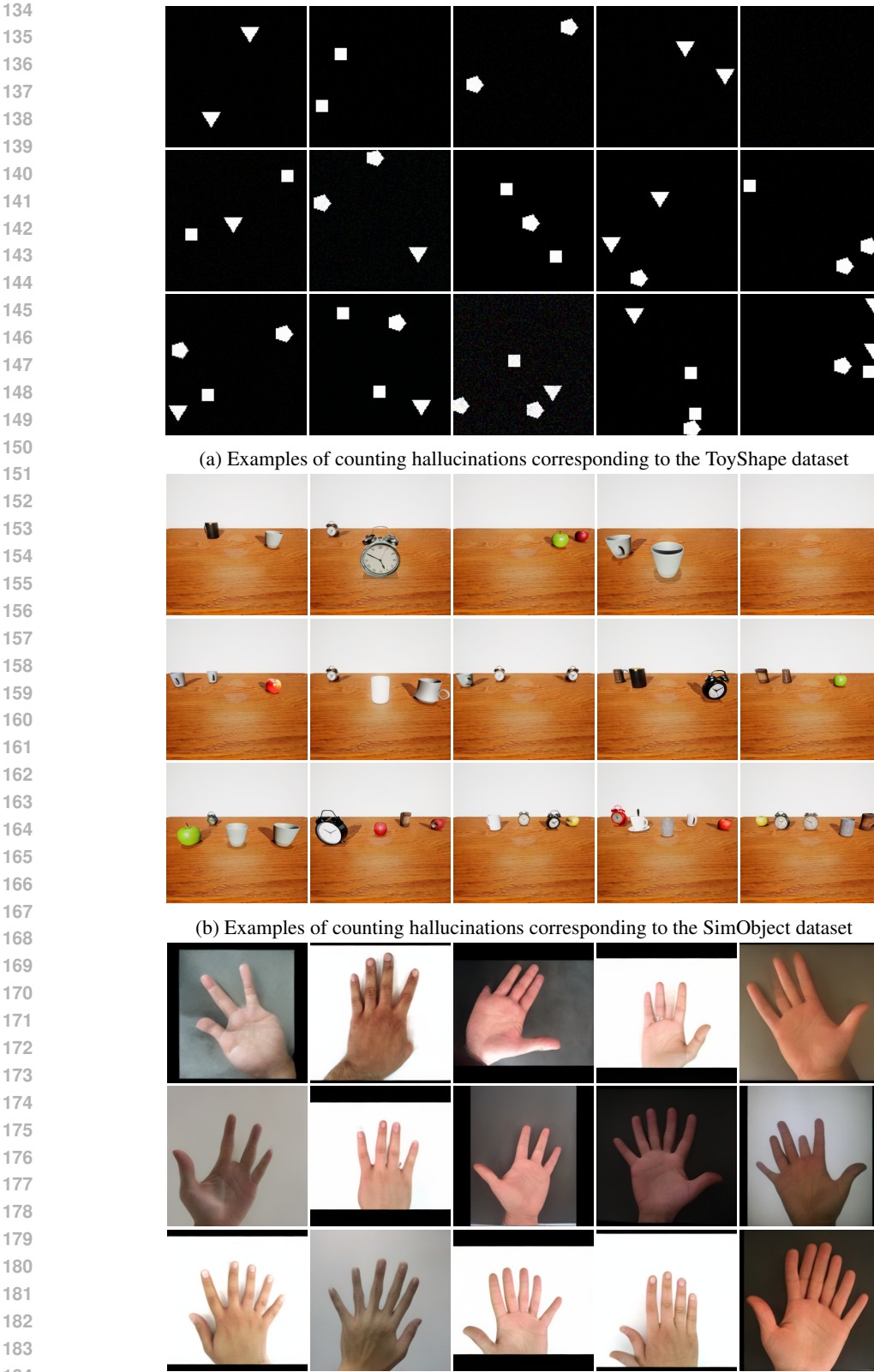

(a) Examples of counting hallucinations corresponding to the ToyShape dataset

(b) Examples of counting hallucinations corresponding to the SimObject dataset

(c) Examples of counting hallucinations corresponding to the RealHand dataset

Figure A2: Examples of counting hallucinations corresponding to the ToyShape, SimObject, and RealHand datasets.

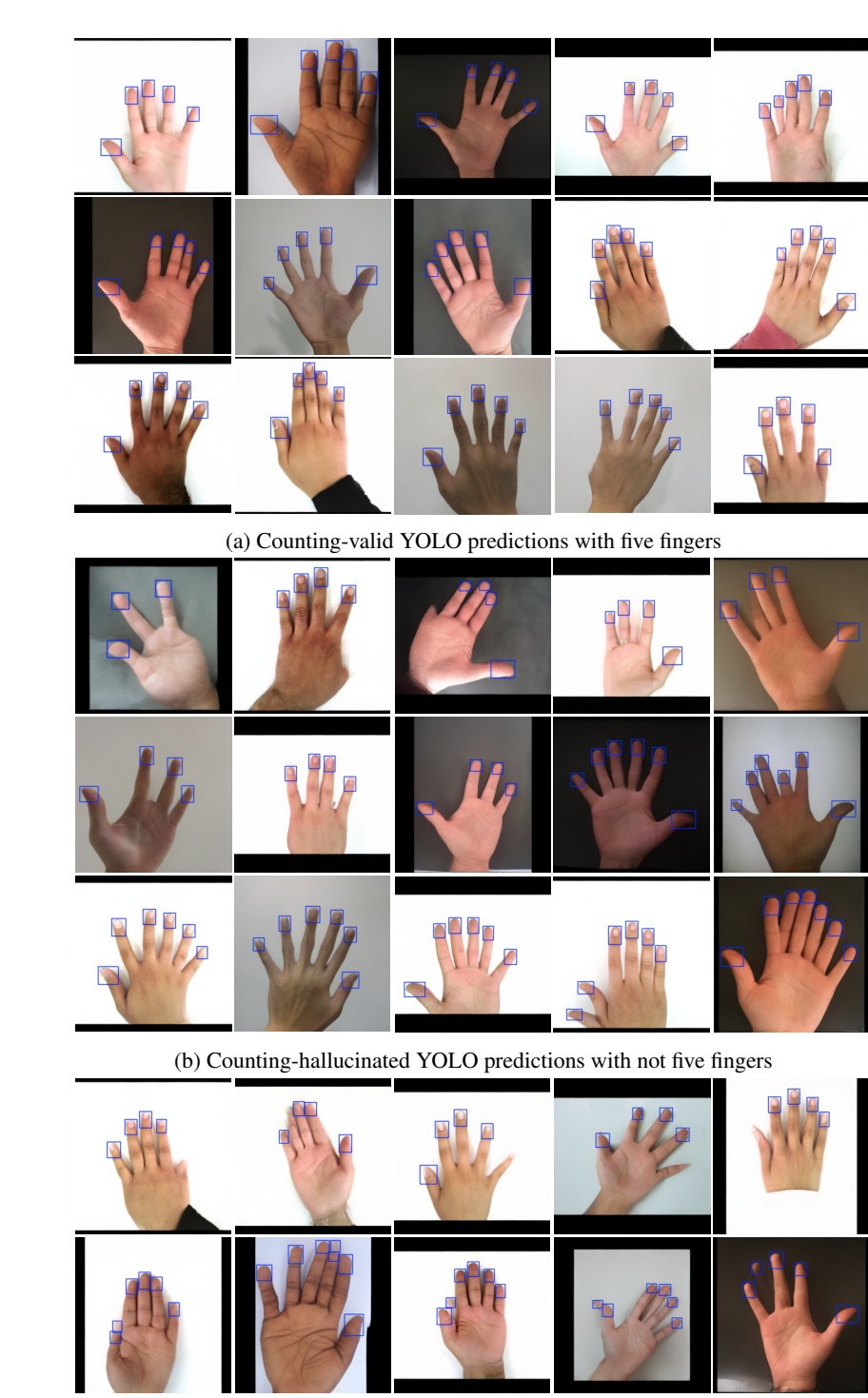

(a) Counting-valid YOLO predictions with five fingers

(b) Counting-hallucinated YOLO predictions with not five fingers

(c) Rare edge cases where YOLO predictions fail

Figure A3: Examples of YOLO predictions on generated RealHand images. (a) Counting-valid samples with five fingertip bounding boxes correctly predicted by YOLO (including cases with two thumbs). (b) Counting-hallucinated samples with incorrect fingertip count predictions (e.g., 3, 4, 6, or 7 detected fingertips). (c) Several rare edge cases where the counting-ready indicator fails to filter out and YOLO struggles to handle, such as blurred, incomplete, or anatomically disconnected fingers.

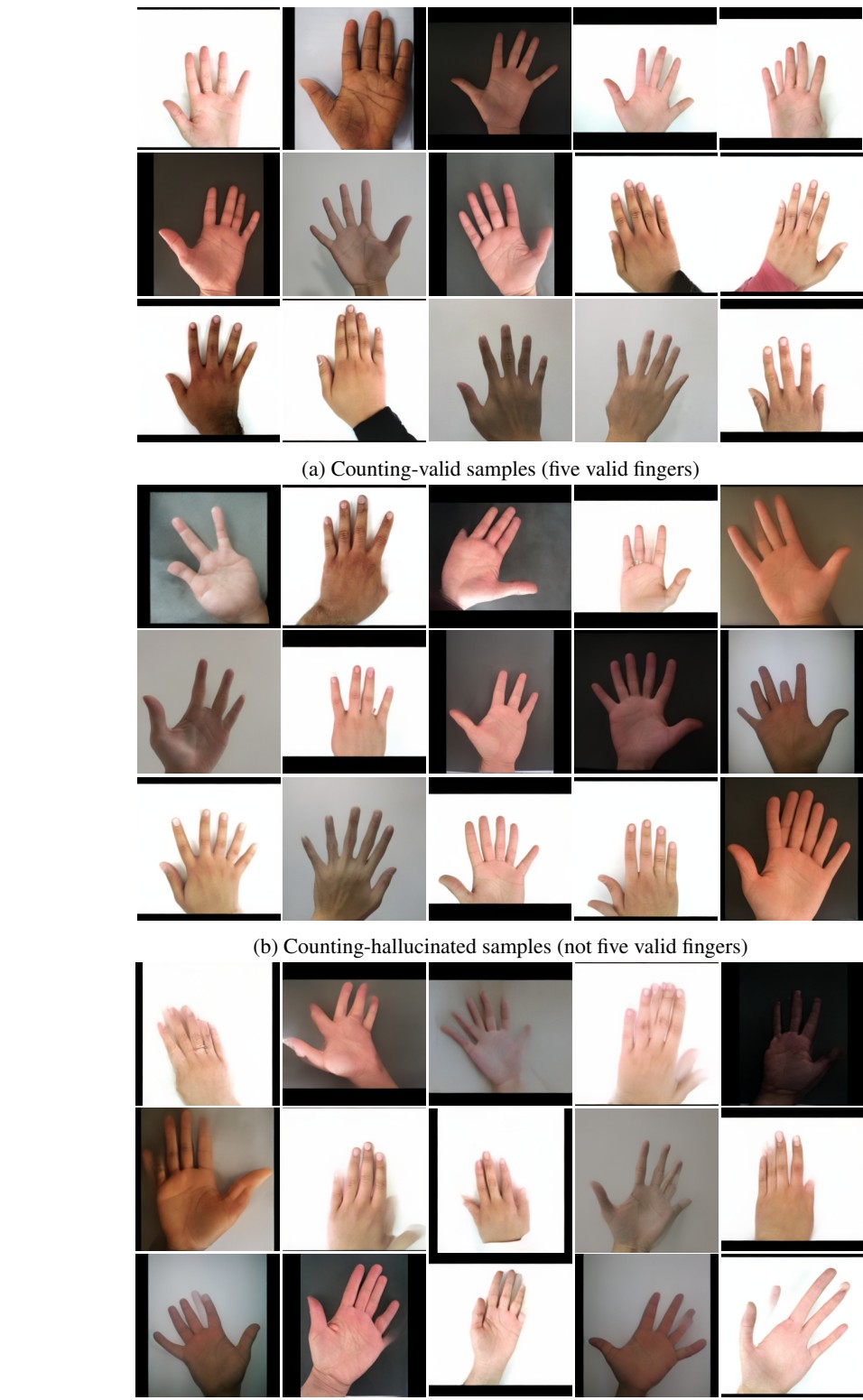

(a) Counting-valid samples (five valid fingers)

(b) Counting-hallucinated samples (not five valid fingers)

(c) Non-counting failure samples (unsuitable for counting)

Figure A4: Examples of three classes of generated hand images. (a) Counting-valid: images with five clearly discernible fingers. (b) Counting-hallucinated: images with an incorrect number of valid fingers (e.g., 4 or 6). (c) Non-counting failures: images unsuitable for counting due to severe visual artifacts, such as distorted, cracked, partially visible, or floating fingers.

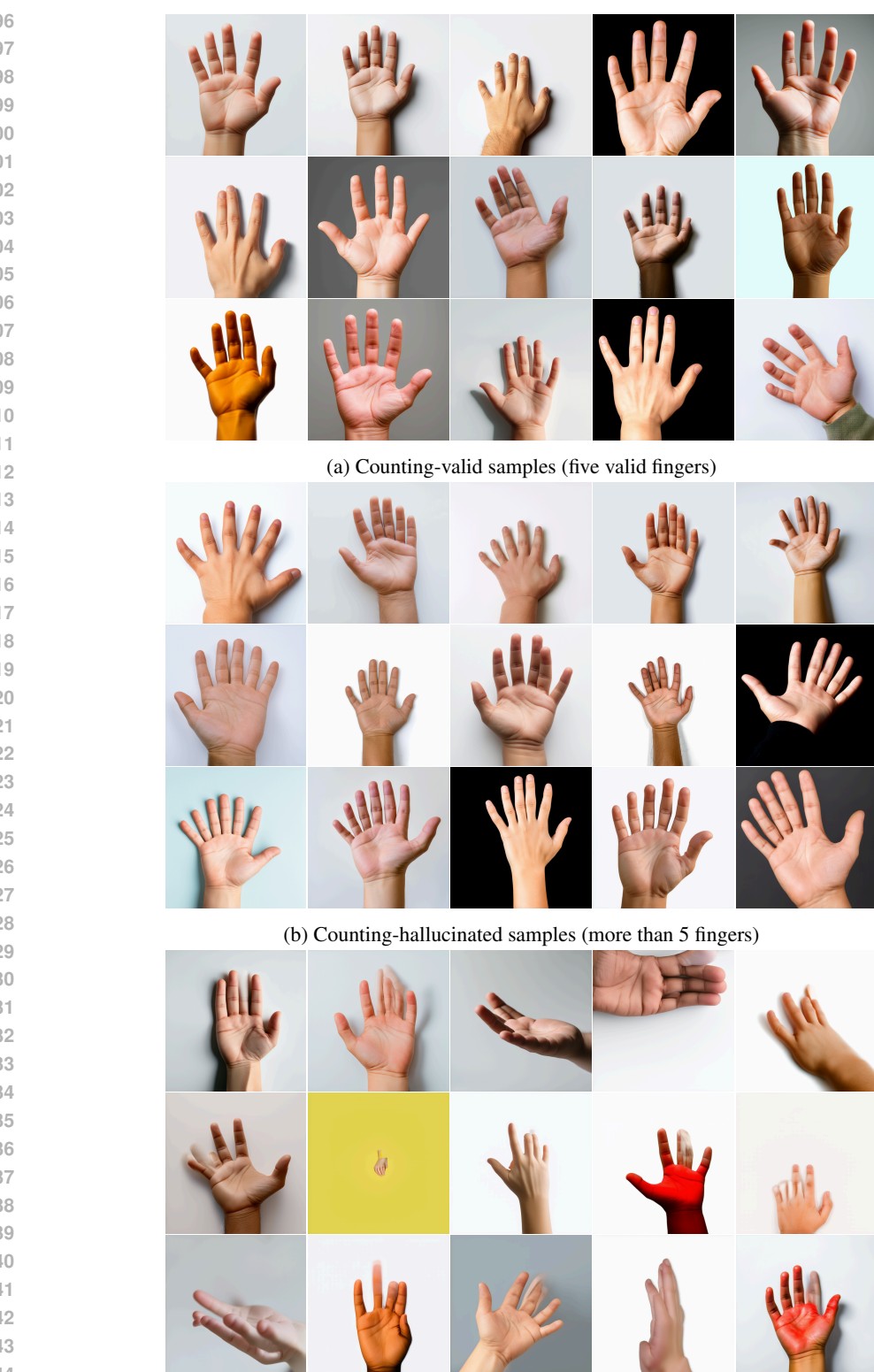

(a) Counting-valid samples (five valid fingers)

(b) Counting-hallucinated samples (more than 5 fingers)

(c) Non-counting failure samples (unsuitable for counting)

Figure A5: Examples of three classes of generated hand images using pretrained Stable-Diffusion-3.5-Medium (PTI-Hand). (a) Counting-valid: images with five clearly discernible fingers. (b) Counting-hallucinated: images with more than five valid fingers (e.g., 6 or 7). (c) Non-counting failures: images unsuitable for counting due to severe visual artifacts, such as distorted, cracked, partially visible, or floating fingers.