# OpenReview forum: "Counting Hallucinations in Diffusion Models"
_ICLR.cc/2026/Conference — Submitted to ICLR 2026_

### Official Review · Reviewer_7KMg · 2025-10-27

**Soundness:** 4
**Presentation:** 3
**Contribution:** 1
**Rating:** 2
**Confidence:** 4

**Summary:**

The paper introduces a framework for quantitatively measuring a specific type of error in image diffusion models called counting hallucinations: cases where models generate an incorrect number of objects (e.g., a hand with missing fingers). The authors build a dataset suite called CountHalluSet and analyze how sampling conditions like solver type, ODE order, number of steps, and initial noise affect hallucination rates. They find that standard quality metrics like FID fail to capture these errors and propose a simple mitigation method called joint-diffusion models to reduce hallucination.

**Strengths:**

1. Hallucination has long been difficult to quantify in generative models. This paper makes progress by isolating and rigorously evaluating one facet of hallucination (counting hallucination) where the number of generated objects is different from the ground truth.
2. The experimental setup, evaluation metrics, and analyses are clearly described and well controlled, which enhances reproducibility and robustness of the results.

**Weaknesses:**

1. The paper does not evaluate state-of-the-art diffusion models (e.g., FLUX, Stable Diffusion), but instead relies on models trained or fine-tuned on relatively small datasets. As a result, it remains unclear whether CountHalluSet meaningfully captures hallucination behavior in modern large-scale diffusion models, or whether the reported effects of sampling parameters generalize to them.
2. The study focuses narrowly on counting hallucinations, whereas real-world hallucinations in image generation include missing/wrong subjects, incorrect spatial relationships, geometry, depth, color, and style inconsistencies. This narrow scope limits the paper’s overall contribution and novelty.
3. The only real-image dataset used, RealHand, consists of single-hand images occupying most of the frame. This dataset lacks diversity in object categories and does not test models on multi-object types or cluttered scenes.

**Questions:**

1. The proposed Joint-Diffusion Model involves classifier guidance from a detector at intermediate denoising steps, which may substantially increase inference time. Did the authors benchmark the impact of the additional guidance on wall-clock generation time?

---

> ### Author Response · Authors · 2025-12-03
>
> We thank the reviewer for the valuable suggestions.
>
> > Q1: Limited scope.
> The study focuses narrowly on counting hallucinations, whereas real-world hallucinations in image generation include missing/wrong subjects, incorrect spatial relationships, geometry, depth, color, and style inconsistencies. This narrow scope limits the paper’s overall contribution and novelty.
>
> __R1__: We respectfully disagree with this concern.
>
> Our goal in this paper is to take the first systematic step toward quantifying counting hallucinations in diffusion models and to establish a feasible, reproducible evaluation protocol for this specific yet fundamental type of hallucination. We explicitly separate counting hallucinations from other non-failure modes, ensuring conceptual clarity and enabling targeted analysis.
>
> Moreover, our additional experiments on large-scale pretrained text-to-image diffusion models demonstrate that the proposed evaluation protocol transfers reliably beyond controlled settings, confirming its extendibility and practical relevance.
>
> Taken together, these contributions, formalizing counting hallucinations, designing a principled evaluation methodology, and validating its robustness across model scales, represent __an important and novel step toward broader and more systematic hallucination quantification in generative models__. All of these are important contributions for the community.
>
> > Q2: The dataset diversity.
> The only real-image dataset used, RealHand, consists of single-hand images occupying most of the frame. This dataset lacks diversity in object categories and does not test models on multi-object types or cluttered scenes.
>
> __R2:__ We thank the reviewer for raising this point. However, we respectfully disagree with the underlying expectation that a dataset designed for isolating counting hallucinations must also cover arbitrary real-world scene diversity.
>
> As discussed in __R5 to Reviewer BJ4C__, our goal in this work is to take a controlled and systematic first step toward quantifying counting hallucinations. RealHand is intentionally focused on single-hand images to remove confounding factors from cluttered scenes, multi-object interactions, and background variation. This controlled setting is necessary to ensure that counting failures can be reliably identified and compared across sampling conditions and solvers.
>
> Nevertheless, to verify that our observations are not restricted to the single-hand setting, we conducted a small preliminary experiment on __Stable-Diffusion-3.5-Medium__ by generating two-hand images using the same evaluation pipeline. The model exhibits a similar trend as reported in Table 1, suggesting that our findings extend beyond the single-hand scenario.
>
> We agree that extending the benchmark to multi-object scenes is __a valuable future direction__ and would further enrich the study. Our protocol is general and can naturally accommodate such extensions.
>
> > Q3: Clarification on JDM.
>
> __R3:__ See __GR1__.

---

### Official Review · Reviewer_BJ4C · 2025-10-28

**Soundness:** 2
**Presentation:** 2
**Contribution:** 2
**Rating:** 4
**Confidence:** 3

**Summary:**

In this paper, the authors discuss the problem of hallucination in image generation by diffusion models, focusing on a specific subtype termed counting hallucination, where generated images contain a physically impossible or dataset-inconsistent number of objects. To quantify this phenomenon, the authors construct a benchmark suite, CountHalluSet, consisting of three datasets of increasing visual complexity, each defined by explicit counting rules. Using these datasets, they systematically analyze how different diffusion sampling conditions (e.g., number of steps, solver type, initial noise) affect counting hallucination. The study further compares hallucination rates with FID, showing that while generation quality metrics like FID correlate with general visual failures, they do not correlate with counting hallucinations, highlighting a gap between perceptual quality and factual correctness. Finally, motivated by these findings, the authors introduce a conceptual Joint Diffusion Model (JDM), which purportedly reduces hallucinations by jointly denoising visual representations and factual constraints in a shared latent space.

**Strengths:**

- The paper is nicely motivated with a relevant and important issue: image generation models often generate images that does not folow phyciscal law.
- The authors focus on counting hallucination and define measureable metrics, makes the analysis quantifiable.
- The proposed CountHalluSet provides a controlled progression of datasets from synthetic toy datasets to real-world datsets, enabling systematic study of hallucination under increasing realism.
- The paper explores some important dimensions of diffusion models including solver types, integration order, and step counts influence hallucination rates.
- Section 5.4 propose interesting future work direction on how to reducte hallucination can insipre future work on constraint-aware diffusion generation.

**Weaknesses:**

Overall, the scope of the studied problem seems limited. The paper focuses narrowly on counting hallucinations but offers little explanation of why diffusion models shows counting hallucination or how diffusion dynamics contribute. The proposed dataset suite is mostly synthetic with arbirtray counting rules. The proposed Joint Diffusion Model (JDM) in Section 5.5 is an interesting idea but discussed too briefly and without sufficient methodological detail.

- Paper writing lack clarity on some experiments. For example, there is no clear definition on the NCFR mentioned in Table 1.It is unclear what hat constitutes a “non-counting hallucination” and how such samples are detected.
- Similarly, there is no clear explanation for the proposed JDM in table 3. Whereas the motivation makes sense and seems promising, the paper lacks discussion on what techniques and architecture was adopted for Table 3.
- The paper motivates with the problem of hallucination as violations of physical laws or semantic consistency. However, the toy datasets primarily enforce arbitrary counting rules rather than true physical or causal constraints. It is questionable how representitive the datasets are for real image generation counting constraints. This can also be confirmed with the inconsistent trend between the datasets in the result section.

**Questions:**

- How exactly are NCFR and TFT measured?
- Can you provide mored detail on the JDM in section 5.5?
- The motivation mentions “violations of physical laws.” How do the synthetic datasets (ToyShape, SimObject) reflect such physical constraints rather than dataset-specific counting rules?

---

> ### Author Response · Authors · 2025-12-03
>
> We thank the reviewer for the insightful and constructive feedback.
>
> > Q1: Clarfication on Table 1.
> Paper writing lack clarity on some experiments. For example, there is no clear definition on the NCFR mentioned in Table 1. It is unclear what hat constitutes a “non-counting hallucination” and how such samples are detected.
>
> __R1:__ NCFR measures failure cases captured by the counting-ready indicator (see the revised caption of Table 1).
>
> > Q2: Generalization to real image generation counting constraints.
> The paper motivates with the problem of hallucination as violations of physical laws or semantic consistency. However, the toy datasets primarily enforce arbitrary counting rules rather than true physical or causal constraints. It is questionable how representitive the datasets are for real image generation counting constraints. This can also be confirmed with the inconsistent trend between the datasets in the result section.
>
> __R2:__ We respectfully disagree with this concern.
>
> First, our goal is to take a first systematic step toward quantifying hallucinations in diffusion models, and the three datasets are not arbitrary. Instead, they are deliberately designed to isolate counting hallucinations under progressively richer physical and semantic structures, from geometric primitives (ToyShape), to physically plausible 3D objects (SimObject), and finally to real human hands with anatomical constraints (RealHand). Importantly, __increasing the complexity of counting constraints does not alter one of our central findings__: the correlation between counting hallucination rate (CHR) and perceptual metrics remains consistently weak, and this holds in a fully controlled and reproducible setting.
>
> Second, the claim of “inconsistent trends” is not supported by the quantitative results. After adding the distance correlation metric, we observe a monotonically decreasing pattern: 0.82 (SimObject) → 0.74 (RealHand) → 0.47 (PTI-Hand), indicating that as generative content becomes more complex, the association between CHR and FID naturally weakens. This trend reflects inherent dataset complexity, not inconsistency.
>
> > Q3: Explanation on why diffusion models shows counting hallucination.
> The paper focuses narrowly on counting hallucinations but offers little explanation of why diffusion models shows counting hallucination or how diffusion dynamics contribute.
>
> __R3:__ Our goal in this paper is to take the first systematic step toward quantifying counting hallucinations in diffusion models; therefore, providing a complete theoretical explanation of why diffusion models exhibit such errors is beyond the scope of this work. We have added a brief discussion in Appendix A1–A3 and highlight it as an important direction for future research.
>
> That said, counting hallucinations can arise from multiple sources within the diffusion process. In particular, diffusion models accumulate both solver error and model estimation error across timesteps. Even though increasing sampling steps improves solver accuracy in principle, in practice, model estimation error at each denoising step, together with the mismatch between training and inference noise schedules, can amplify structural inconsistencies such as over- or under-counting.
>
> __These errors are tightly entangled and difficult to isolate theoretically, which motivated our empirical strategy__: we systematically quantify counting hallucinations across diverse solvers and sampling conditions to reveal how different numerical and modeling factors influence their occurrence.
>
> > Q4: How exactly are NCFR and TFT measured?
>
> __R4__: NCFR measures failure cases captured by the counting-ready indicator (see the revised caption of Table 1).
>
> > Q5: Motivation of violations of physical laws.
> The motivation mentions “violations of physical laws.” How do the synthetic datasets (ToyShape, SimObject) reflect such physical constraints rather than dataset-specific counting rules?
>
> __R5:__ We would like to clarify a misunderstanding regarding the role of “physical laws” in our motivation. __Our paper does not claim that ToyShape or SimObject explicitly encode full physical laws__ such as mechanics or optics. Rather, we motivate hallucinations as violations of factual constraints, including counting constraints, commonsense factual constraints, and physics-based constraints, among which counting provides the most objective, observable, and feasible starting point.
>
> Given the complexity and subjectivity involved in modeling richer physical or commonsense constraints, this work intentionally takes the first systematic step by focusing on counting, where __ground-truth is unambiguous and evaluation is reproducible__.
>
> > Q6: Clarfication on Table 1.
> Paper writing lack clarity on some experiments. For example, there is no clear definition on the NCFR mentioned in Table 1. It is unclear what hat constitutes a “non-counting hallucination” and how such samples are detected.
>
> __R6:__ See __R4__.

---

### Official Review · Reviewer_a9sr · 2025-10-29

**Soundness:** 3
**Presentation:** 3
**Contribution:** 1
**Rating:** 4
**Confidence:** 4

**Summary:**

The paper investigates the problem of counting hallucinations. To this end, the authors construct CountHalluSet, a benchmark comprising three components: ToyShape, SimObject, and RealHand, ranging from simple synthetic shapes to real-world hand images. Using this dataset, the paper examines how factors such as solver type, sampling steps, and initial noise influence counting hallucinations. Interestingly, the study reveals that counting hallucinations are not directly correlated with overall generation quality. Finally, the authors propose a method to mitigate these hallucinations, contributing a practical solution to the issue.

**Strengths:**

1. This paper is the first to systematically study counting hallucinations in diffusion models. In large-scale text-to-image diffusion models, counting hallucinations present a genuine challenge. Therefore, this work has the potential to provide a systematic and quantitative understanding of counting hallucinations in such models.

2. The paper is well-structured: the definition of counting hallucination is clearly articulated, the experimental setup is well explained, and the proposed dataset is thoughtfully constructed. Moreover, the experimental findings, such as the observation that counting hallucinations are not intrinsically related to generation quality, are both interesting for the research community.

**Weaknesses:**

1. The paper claims a contribution in proposing a method to reduce counting hallucinations. However, the description of this method lacks sufficient detail. For instance, what exactly are the "joint-diffusion models (JDM)" mentioned in line 464? More clarification and elaboration are necessary.

2. There is a noticeable gap between the contributions of this paper and the issues that people care about in practice. Specifically, the authors evaluate primarily on synthetic datasets and a small-scale hand dataset to draw conclusions. It is unclear how these findings generalize to large-scale models, which is the community’s main concern. I recommend the following additional experiments to strengthen the paper:

      a. Study counting hallucinations in large-scale text-to-image diffusion models such as Stable Diffusion or Flux. For example, generate images containing hands and evaluate how solver type, sampling steps, and initial noise affect counting hallucinations. Additionally, investigate the relationship between generation quality and counting hallucinations in these settings.

      b. Evaluate the performance of the proposed joint-diffusion models on large-scale text-to-image diffusion models, to assess their practical effectiveness.

If the authors address these concerns, I would be inclined to raise my score.

**Questions:**

1. How to obtain the "ground-truth initial noise" in line 372.
2. How many samples are generated to evaluate results in Table 1 and Table 2?

---

> ### Author Response · Authors · 2025-12-03
>
> We thank the reviewer for the valuable suggestions.
>
> >Q1: Practical effectiveness of JDM on large-scale text-to-image diffusion models.
>
> __R1:__ As discuessed in __GR1__ and Appendix A9, JDM is novel training paradigm for improving factual generation in diffusion models rather than as a superior inference strategy. Essentially, extending a pretrained text-to-image model with additional mask channels __disrupts the manifold geometry of the pretrained latent space and underlines the original denoising distribution__, making direct finetuning such model nontrivial.
>
> Despite this challenge, we still observe promising reductions in counting hallucinations when applying JDM to large-scale pretrained diffusion models under carefully controlled finetuning conditions. In practice, we find that jointly aligning the extended latent space in JDM with the __highly rigid latent geometry of large pretrained models__ is challenging under the limited scale of RealHand. To strike a practical balance, we adopt a two-stage finetuning strategy: we first finetune a Stable-Diffusion-1.5 model on RealHand, then freeze all intermediate UNet layers and finetune only the input and output layers with extended channels. As shown in Table A3, __even under this highly constrained finetuning setup, the joint-diffusion paradigm consistently yields lower counting hallucination rates__ at 25 and 50 sampling steps across both solvers.
>
> We believe that exploring its behavior and performance __under true large-scale training__ remains an important and compelling avenue for future research.
>
> > Q2: How to obtain the "ground-truth initial noise" in line 372.
>
> __R2:__ The ground-truth initial noise in Table 1 is obtained via the standard forward diffusion process with 1000 timesteps. For fairness and reproducibility, the same noise vectors are reused across all evaluations for each seed.
>
> > Q3: How many samples are generated to evaluate results in Table 1 and Table 2?
>
> __R3__: Unless noted otherwise, the number of generated samples matches the number of training samples in each evaluation (see Section 5.1).
>
> > Q4: Generalization to Large-Scale Pretrained Text-to-Image Models.
> There is a noticeable gap between the contributions of this paper and the issues that people care about in practice. Specifically, the authors evaluate primarily on synthetic datasets and a small-scale hand dataset to draw conclusions. It is unclear how these findings generalize to large-scale models, which is the community’s main concern. I recommend the following additional experiments to strengthen the paper:
> a) Study counting hallucinations in large-scale text-to-image diffusion models such as Stable Diffusion or Flux. For example, generate images containing hands and evaluate how solver type, sampling steps, and initial noise affect counting hallucinations. Additionally, investigate the relationship between generation quality and counting hallucinations in these settings.
>
> __R4:__ See __GR2__.

---

### Official Review · Reviewer_N6Cv · 2025-11-01

**Soundness:** 3
**Presentation:** 3
**Contribution:** 2
**Rating:** 4
**Confidence:** 4

**Summary:**

This paper builds on top of the hallucination investigation by Aithal et. al. by introducing a benchmark suite and a complementary method that allows quantifying “counting hallucinations” in diffusion models. These are cases where diffusion models generate incorrect number of objects (such as extra fingers). Three datasets are used with different complexity: ToyShape, SimObject, and RealHand.

The key findings of the paper are that: (1) standard image quality metrics like FID fail to measure counting hallucinations; (2) While increasing sampling steps reduces hallucinations in the simple datasets, it often increases them in complex ones; and (3) a Joint Diffusion Model (JDM) method helps reduce hallucinations across the board.

**Strengths:**

1. The problem has been studied in a very clean way. The authors articulate how counting hallucinations is very well positioned because of its quantifiable nature. On top of that, the three levels of complexity allow for a very clear comparison.
2. JDM provides a promising initial direction for enforcing factual constraints in generative models.
3. The lack of correlation of count hallucinations with FID reflects how perceptual and factual quality may not always agree. This is a nice point.
4. Really like the pre-filtering with counting-ready indicator! This makes so much sense.

**Weaknesses:**

1. The experiments on JDM are rather short. I would have liked to see a broader discussion of the same across datasets.
2. How does JDM impact FID? How is JDM implemented? The section is very underdevleoped and not conference ready.
3. Since the FID correlations change so much across solvers and datasets, it really calls for more experiments: either more finetuning runs, or more datasets, or more solvers. We need more evidence before making any statistically significant claim

**Questions:**

1. I do not think I followed how NCFR is computed. What are the other failures for each dataset?
2. Which dataset is Table 3 on?

---

> ### Author Response · Authors · 2025-12-03
>
> We thank the reviewer for the insightful and constructive feedback.
>
> > Q1: More correlation studies.
> Since the FID correlations change so much across solvers and datasets, it really calls for more experiments: either more finetuning runs, or more datasets, or more solvers. We need more evidence before making any statistically significant claim.
>
> __R1:__ To strengthen the correlation analysis, we have substantially expanded our experiments in several directions.
>
> First, we increased the coverage of sampling configurations by adding ODE-solver results at additional step counts (25, 50, 100 → 25, 50, 100, 200, 500, 1000), and incorporated all results into an updated Table 2.
>
> Second, we introduced an additional and widely used perceptual metric, __MUSIQ__, to complement FID and provide a more comprehensive assessment of perceptual quality.
>
> Third, to further validate the robustness of our findings, we included correlation results from __a large-scale pretrained text-to-image diffusion model__, demonstrating that the observed trends persist beyond the RealHand setting.
>
> Together, these extended experiments provide stronger empirical evidence supporting the claims made in the paper.
>
> > Q2: Clarification on NCFR
> I do not think I followed how NCFR is computed.
>
> __R2:__ NCFR measures failure cases captured by the counting-ready indicator (see the revised caption of Table 1).
>
> > Q3: Clarification on other failures for each dataset.
> What are the other failures for each dataset?
>
> __R3:__ In CountHalluSet, only the RealHand dataset exhibits non-counting failure cases. This is because the synthetic datasets (ToyShape and SimObject) are structurally simple and visually clean, making severe degradations or implausible artifacts extremely rare (see Figure 2).
>
> > Q4: Clarification on Table 3.
> Which dataset is Table 3 on?
>
> __R4:__ RealHand.
>
> > Q5: Clarification on JDM.
> The paper claims a contribution in proposing a method to reduce counting hallucinations. However, the description of this method lacks sufficient detail. For instance, what exactly are the "joint-diffusion models (JDM)" mentioned in line 464? More clarification and elaboration are necessary.
>
> __R5:__ See __GR1__.

---

### Author Response · Authors · 2025-12-03
**Overview of Manuscript Revisions**

Thank you for all reviewers’ constructive feedback.

In response to the concerns raised, particularly regarding the generalizability of our RealHand findings to large-scale pretrained text-to-image models and implementation details of JDM, we have undertaken substantial revisions and clarifications throughout the manuscript. Below, we summarize the key updates.

**Summary of Revisions and Additions**

- **Revised Contributions (Sec. 1):** We refined the presentation of our main contributions to improve clarity and highlight the scope of this work.
- **Updated Observations (Sec. 5):** We streamlined and clarified the empirical observations to make the findings more accessible.
- **Additional Experiments (Tab. 2 & Fig. 3):** We incorporated new experimental results evaluating counting hallucinations in large-scale pretrained text-to-image models, enabling direct comparison with the RealHand setting.
- **New Section on Generalization (Sec. 5.4):** We added a dedicated section discussing how the trends observed in RealHand transfer to pretrained text-to-image models, addressing concerns about external validity.
- **Expanded JDM Implementation Details:** We added detailed descriptions of JDM, including a new diagram (Fig. 4), for clearer methodological understanding.
- **New Theoretical Discussions (App. A1–A3):** We introduced theoretical analyses outlining potential and inherent factors in diffusion models that influence counting hallucinations.
- **Additional Representation Examples (App. A5–A7):** We included some representative qualitative examples.
- **Extended Pretrained T2I Model Details (App. A8):** We added further implementation details for experiments involving large-scale pretrained models.
- **Additional JDM Results and Discussion (App. A9):** We provided more comprehensive results, analyses, and discussions related to JDM.

---

### Author Response · Authors · 2025-12-03
**General Response**

We address the general questions point by point as follows.

__GR1__: __Implementation Details of JDM__

We thank all reviewers for highlighting the need for a clearer and more comprehensive description of our Joint-Diffusion Model (JDM). In response, we have substantially added implementation details and experimental analysis in the revised manuscript.

__(1) Added architectural clarification and implementation details.__
We added Figure 4 and a detailed explanation of the JDM architecture in Section 5.5 and Appendix A9. Specifically, JDM performs a channel-wise concatenation between the latent representation of a hand mask and that of the corresponding RGB image, as illustrated in Figure 4. This joint modeling encourages anatomically plausible hand poses and enforces correct finger topology during generation. We further describe how the UNet backbone, pretrained on CelebA-HQ following the RealHand setup, was finetuned using RealHand images and their SAM-2–extracted masks.

__(2) Added performance analyses of FID for JDM.__
Reviewer __N6Cv__ raised the question of the impact of JDM on perceptual quality. We incorporated FID results into Table 3 and expanded the discussion in Section 5.5.

On the one hand, JDM consistently reduces both counting hallucination rate (CHR) and non-counting failure rate (NCFR) across all sampling conditions, demonstrating its effectiveness in generating structurally intricate objects that are particularly susceptible to counting ambiguities.

On the other hand, __JDM yields slightly higher FID__. We attribute this to the increased denoising burden introduced by jointly modeling two modalities under a small-scale training regime, which forces the model to trade off perceptual quality for improved structural realism.

__(3) Addressing inference-time overhead concerns.__
Reviewer __7KMg__ raised the question of computational overhead. As clarified in the revision, our JDM design does not introduce classifier-based guidance during intermediate steps. Instead, __it is a novel training paradigm__. It modifies only the input and output channels of the UNet while keeping all intermediate layers unchanged. Consequently, JDM adds __negligible inference-time overhead__ relative to the vanilla LDM. We have made this explicit in Appendix A9.

---

__GR2__: __Generalization to Large-Scale Pretrained Text-to-Image Models__

Reviewr __a9sr__ and __7KMg__ raised concerns regarding whether the findings from RealHand generalize to large-scale pretrained text-to-image diffusion models.

Firstly, we clarify a fundamental challenge. __the definition of hallucination__ assumes that the model generates patterns that __never appear in its training data__. While this assumption is verifiable for our controlled RealHand setting, it is intractable for large-scale pretrained diffusion models, where the full training distribution is unavailable. This is __the primary reason why we did not initially evaluate counting hallucinations on pretrained text-to-image models__ such as Stable Diffusion.

Secondly, __the inherent faithfulness inconsistencies in text-to-image models__ make the evaluation of counting hallucinations more tricky.

Nevertheless, to enable meaningful quantification despite these limitations, we adopt a __conservative and empirically justified criterion__. We treat any generated hand image containing __six or more fingers__ as a counting hallucination. Although real images may show fewer than five visible fingers due to occlusion, articulation, or camera viewpoint, it is highly implausible for real data to contain hands with six or more fingers. Thus, a finger count of $\ge$ 6 serves as a reliable indicator of counting hallucination in pretrained models.

To address reviewers’ concerns, we have added Section 5.4 and Appendix A8, and updated Table 2 and Figure 3 with extensive experiments on __Stable-Diffusion-3.5-Medium__. These experiments leverage 10 diverse hand-generation prompts while avoiding additional faithfulness-related ambiguities, and are evaluated using the counting-ready indicator and counting model introduced in Section 4.2.3.

As shown in Figure 3, the way solver type and sampling steps affect counting hallucinations in large pretrained diffusion models __aligns closely__ with the trends revealed by our controlled RealHand experiments. This consistency highlights the validity of the findings presented in Section 5.2 and underscores the broader applicability of our evaluation protocol. Correlation results in Table 2 further support this conclusion. In specific, weak correlations between CHR and perceptual metrics and strong correlations between NCFR and perceptual metrics are also observed.

Together, these results demonstrate the __robustness and generality of our findings on RealHand__ and show that the proposed evaluation protocol transfers reliably to pretrained text-to-image models.

---

### Author Response · Authors · 2025-12-03

Dear Area Chairs,

We sincerely regret the recent information leak. Despite this unfortunate situation, we respectfully ask the AC to carefully consider our revised manuscript and rebuttal. We believe that the additional experiments, clarifications, and revisions we have provided adequately address the majority of the reviewers’ concerns. Since the reviewers are no longer able to respond at this stage, we would be happy to provide any further clarification if needed.

Best,
Authors

---

### Meta-Review · Area_Chair_DdWN · 2026-01-11

**Summary:**

Reviews began as generally negative, citing a number of concerns about the scope, diversity, and impact of the proposed benchmark, as well as issues with missing details in the proposed method (JDM). Some of these details have been addressed, but many remain outstanding as disagreements between the authors and reviewers about the intended impact of the paper.

**Reviewer Concerns:**

Reviewers raised the following concerns:

(At least partly) Addressed:
- Not enough details were provided about the JDM (how it works, architecture, etc). Additionally, concerns were raised about the computational overhead of the JDM. These were addressed by the authors in their response and in a revision of the manuscript.
- Doubts that the small datasets generalize to large-scale models like Stable Diffusion or FLUX. In response, the authors provided an explanation of why this wasn't provided, and some analysis of a medium-scale variant of Stable Diffusion.
- Reviewers express concerns about whether counting rules based on synthetic datasets in the paper are representative of models following physical laws.

Outstanding:
- Reviewers comment that hallucinations can not always be counted, and this paradigm does not cover other types of model errors, such as spatial incoherence or semantic errors.
- Reviewers complain that the datasets provided do not seem diverse or representative of real-world content (e.g., RealHand is limited to a single hand and is not in a real scene-like setting).

**Reviewer Scores:**

Reviewer scores were initially negative (4,4,4,2), and the rebuttals did not suggest to me that the scores may have changed considerably. I expect that one or two borderline reviewers may have increased their scores to 5 following more discussion, but the paper will likely have remained with borderline-negative scores, e.g., (5,4,4,2) or (5,5,4,2).

---

### Decision · Program_Chairs · 2026-01-26

Reject